# Nasal Spray Disinfectant for Respiratory Infections Based on Functionalized Silver Nanoparticles: A Physicochemical and Docking Approach

**DOI:** 10.3390/nano15070533

**Published:** 2025-03-31

**Authors:** Benjamín Valdez-Salas, Jorge Salvador-Carlos, Ernesto Valdez-Salas, Ernesto Beltrán-Partida, Jhonathan Castillo-Saenz, Mario Curiel-Álvarez, Daniel Gonzalez-Mendoza, Nelson Cheng

**Affiliations:** 1Core Facilities of Chemistry and Advanced Materials, Instituto de Ingeniería, Universidad Autónoma de Baja California, Calle de La Normal S/N and Boulevard Benito Juárez, Mexicali 21100, Baja California, Mexico; benval@uabc.edu.mx (B.V.-S.); beltrane@uabc.edu.mx (E.B.-P.); jhonathan.saenz@uabc.edu.mx (J.C.-S.); mcuriel@uabc.edu.mx (M.C.-Á.); 2Centro Médico Ixchel, Av. Nicolás Bravo 270, Mexicali 21000, Baja California, Mexico; valdez68ernesto@gmail.com; 3Instituto de Ciencias Agrícolas, Universidad Autónoma de Baja California, Carretera a Delta s/n, Ejido Nuevo Leon, Mexicali 21705, Baja California, Mexico; danielg@uabc.edu.mx; 4Magna International Pte Ltd., 10 H Enterprise Road, Singapore 629834, Singapore; nelsoncheng@magnachem.com.sg

**Keywords:** nasal spray disinfectant, SARS-CoV-2, respiratory diseases, tannic acid, sodium citrate, silver nanoparticles

## Abstract

Respiratory diseases have presented a remarkable challenge during modern history, contributing to important pandemics. The scientific community has focused its efforts on developing vaccines and blocking the transmission of viruses through the respiratory tract. In this study, we propose the use of stable silver nanoparticles (AgNPs) functionalized with tannic acid (TA) and sodium citrate (SC) as a nasal spray disinfectant (NSD). The non-ionic ethoxylated surfactant Tween 80 (T80) was added to enhance the wetting effect on nasal and oral tissues following spray application. We analyzed the physicochemical properties of the AgNPs and the NSD, including zeta potential, polarity, morphology, composition, particle size, and distribution. The results indicated spherical AgNPs ranging from 3 to 5 nm, stabilized by TA-SC. The addition of T80 resulted in particles with negative polarity, high stability, and improved coverage area. Furthermore, the colloidal stability was monitored over one year, showing no signs of degradation or precipitation. Interestingly, the interaction between the capped AgNP complex, the spike protein, and ACE2 was studied by molecular docking, indicating a strong and thermodynamically favorable complex interaction. These findings hold promise for the development of potential inhibitors, antagonist receptors, Ag-complex agonists (as observed here), and drug development for viral protection.

## 1. Introduction

The nasal, oral, and ocular mucous acts as the main airway route for the SARS-CoV-2 penetration and invasion of the receptor cells [1,2]. Thus, following this initial infection process, viral particles can reach the respiratory system, producing alveolar alterations and subsequent pulmonary collapsing, as described among the literature [3,4,5,6]. Interestingly, it was described as having three phases in the pathogenesis of COVID-19 [7]. The first phase is related to the infection of the ciliary and secretory cells in the nose, resulting in asymptomatic individuals due to a low immune response. In the second phase, COVID-19 spreads to the airways of the respiratory tract, the bronchi, and bronchioles, infecting the ciliary cells. Finally, the third phase showed the infection reaches the alveoli, infecting type II cells and interfering with gas exchange. Consequently, a chain reaction develops, characterized by the accumulation of fluid in the bronchi, which alters the surfactant released by type II pneumocytes, endothelial cells’ detachment, and Acute Respiratory Distress Syndrome [8,9]. Finally, the condition can spread to the digestive, urogenital, circulatory, and nervous systems [10,11,12,13].

Therefore, it is mandatory stopping the passage of external agents since the first contact phase is the gold standard for all the systems used for the prevention of SARS-CoV-2 and other respiratory diseases [14,15]. In addition to the prevention protocols implemented in the pandemic along with vaccination, nasal spray gained relevance as a protection system, with an important role due to its simple and non-invasive application. It has been shown that the nasal route has advantages over traditional drug delivery routes [16]. For example, Xu et al. [17] lists ten distinctive advantages for drug release through this route that encompass its behavior against drugs, such as drugs with poor stability in fluids, permeability, and lipophilicity; shortening of the route, including a large adsorption surface, evasion of gastrointestinal conditions, and a precise route to the blood and brain; and its convenience, being non-invasive for long therapies and easy use.

The relevant advantages offered by nasal administration make it the accessible road to implementing new technologies and equipment for preventing respiratory infections [18,19]. Interestingly, strong antivirals coated or encapsulated with affinity materials to the nasal mucosa have been developed with promising results. For example, N-palmitoyl-N-monomethyl-N,N-dimethyl-N,N,N-trimethyl-6-O glycolchitosan, Iota-carrageenan, and hydroxypropyl methylcellulose are among the antivirals that have been shown to have appropriate physicochemical and antimicrobial characteristics for the prophylaxis of COVID-19 using the nasal route [20,21,22,23]. However, despite all these advantages, the nasal route still represents a challenge for drug release, since antimicrobials must display specific physicochemical characteristics, which are affected by factors from the atomization instrument to the drug itself [24,25]. A potential solution to approach these drawbacks could be the design of nanostructured materials with a controlled particle size, charge, concentration, and surface chemistry [26,27,28,29,30].

A clear example of these nanostructured materials is AgNPs, which have demonstrated broad-spectrum antiviral activity when functionalized with natural products or coated with biocompatible surfactants [31,32,33]. Their antiviral activity is due to stable interactions with viral envelopes, inhibition of viral replication, and destabilization of the viral structure [34,35]. Far more important, though, is how their antiviral behavior is modulated following principles and observed in different viruses, including enveloped viruses, such as those of SARS-CoV-2 variants [36].

Considering the above stated information and the important current efforts required to reduce or eradicate potential viral respiratory infections, here, we propose antiviral, natural-based AgNPs engineered with a non-ionic ethoxylated and biocompatible surfactant coating. We also propose assessing the fine-tuned AgNPs, when used in the nasal mucous airway, for their capability of interacting with different variants of SARS-CoV-2, as determined with molecular docking analyses and physicochemical approaches. The findings demonstrate that this NSD potentially prevents respiratory infections, with affordable and non-invasive administration.

## 2. Materials and Methods

### 2.1. Synthesis of AgNPs

A chemical reduction method was employed to synthesize AgNPs. Initially, tannic acid (TA, 0.025 mM, Sigma-Aldrich, St. Louis, MO, USA) and sodium citrate (SC, 5 mM, Fermont, Ciudad De Mexico, Mexico) were simultaneously dissolved in 90 mL of distilled water and refluxed under magnetic stirring in the absence of light until the solution turned pale yellow. At this stage, 10 mL of a 25 mM AgNO_3_ solution (Fermont, Mexico) was added. The solution’s color changed from pale yellow to brown, indicating the formation of AgNPs. The reflux was then stopped, and the solution was allowed to cool to room temperature. Finally, the nanoparticle solution was filtered using Whatman No. 4 paper and stored at room temperature in amber glass containers in dark conditions for future analysis (Figure 1).

### 2.2. Preparation of Nasal Spray Disinfectant (NSD)

An NSD was prepared by incorporating Tween 80 (T80, 15 mM, Sigma Aldrich, USA) into a colloidal solution of AgNPs. This non-ionic ethoxylated surfactant was added to improve the stability of AgNPs and the wetting properties on nasal and oral mucosa during spray application (Figure 1).

### 2.3. Physicochemical Characterization of AgNPs

#### 2.3.1. UV-Vis Spectroscopy

To corroborate the effective synthesis of AgNPs in the resulting colloidal solutions, UV-Vis analysis (UV2600, Shimadzu, Kyoto, Japan) was performed. For spectral recording, 2.5 mL of the colloidal solution was poured into a quartz cell and scanned from 800 nm to 200 nm with a resolution of 1 nm at room temperature.

#### 2.3.2. Dynamic Light Scattering (DLS)

The size (hydrodynamic diameter), particle distribution, zeta potential, and polarity of the synthesized nanoparticles were evaluated using the DLS technique. The AgNPs were analyzed (1 mL) using a Nanotrack Wave II (Microtrac, North Wales, PA, USA) with a 30 s lag and scan time at room temperature. The refractive index values established for the particle and the aqueous medium were 1.75 and 1.33, respectively.

#### 2.3.3. Scanning Electron Microscopy (SEM) and Dispersive X-Ray Energy (EDX)

The morphology and size of the synthesized nanoparticles were characterized by SEM (LYRA 3 Tescan, Brno, Czech Republic) using 20 kV acceleration voltage and a STEM detector. For the preparation of the samples, 1.5 mL of each colloidal solution was centrifuged three times at 12,500 rpm for 10 min each. The supernatant was removed and the resulting pellet was suspended in distilled water. Finally, the recovered precipitate was deposited on a 300 mesh transmission copper grid with ultra-thin carbon film, dried at room temperature, and analyzed by SEM. Furthermore, chemical analysis was performed by EDX coupled with SEM using 10 kV with a long spot size to set a suitable count per second rate for spectrum collection.

#### 2.3.4. Transmission Electron Microscopy (TEM) and Selected-Area Electron Diffraction (SAED)

The morphology and size of the AgNPs were characterized through transmission electronic microscopy (TEM) by using a JEOL 2010 microscope (JEOL, Tokyo, Japan) operating at an accelerating voltage of 200 kV. The samples were prepared by depositing a 10 µL of AgNPs colloid and NSD into a carbon-coated Cu cell, followed by a drying process at room temperature in a desiccator. The crystalline structure of AgNPs was studied through selected-area electron diffraction (SAED), obtained during the TEM characterization. The crystallographic phases were identified by comparing diffraction patterns obtained by the information provided by the Joint Committee on Powder Diffraction Standards (JCPDS), using the JCPDS 04–0783 crystallographic chart.

#### 2.3.5. Fourier Transform Infrared Spectroscopy (FTIR)

The capping of AgNPs by TA and SC was analyzed by FTIR (PerkinElmer Frontier, PerkinElmer, Waltham, MA, USA). FTIR spectra were recorded using an ATR device in the range of 4000 to 400 cm^−1^ with a resolution of 0.5 cm^−1^ at room temperature.

### 2.4. Cell Culture and Cytotoxicity Test

To analyze the cytotoxic effect of the AgNPs, we performed the MTT (3-(4,5-dimethylthiazol-2-yl)-2,5-diphenyl tetrazolium bromide) viability test, as described previously [37,38]. For the viability test, we used a human osteosarcoma-derived osteoblast-like cell line (MG-63, ATCC CRL-1427), cultured at 1 × 10^4^ cells/mL in complete medium constituting Dulbecco’s modified Eagle medium (DMEM, Thermo Fisher Scientific, Waltham, MA, USA) supplemented with 10% heat-inactivated fetal bovine serum (FBS, Thermo Fisher Scientific, Waltham, MA, USA) and 100 units/mL of penicillin–streptomycin (Thermo Fisher Scientific, Waltham, MA, USA) in separate 96-well flat-bottomed polystyrene culture plates (Corning, Midland, MI, USA) at 37 °C in a humidified 5% CO_2_ incubator for 24 h. Then, the cells were washed thrice for 5 min with warm 1 × phosphate-buffered saline (PBS) and incubated for 24 h with serial dilutions (62.5 µg/mL–0.4 µg/mL) of the experimental AgNPs suspended in complete medium. The AgNPs doses were discarded, and the cells were carefully washed thrice with warm PBS, to remove any artifact present in the cultures. Afterwards, the MTT protocol was carried out by adding 100 µL of MTT (5 mg/mL, Sigma Aldrich, St. Louis, MO, USA) to each well of the cultured 96-well plate and incubated at 37 °C in a humidified 5% CO_2_ incubator for 3 h. The resulting formazan crystals were dissolved by removing the remaining medium containing MTT. The plate was placed in an orbital shaker at 140 rpm with 200 µL of dimethyl sulfoxide (Sigma Aldrich, USA) for 20 min. Next, the optical density (O.D.) of the dissolved crystals was measured at 590 nm using a microplate reader (Thermoskan, Thermo Fisher Scientific, USA). The baseline controls were measured in a series of culture wells containing AgNPs at the experimental dilutions and prepared for MTT without MG63 as de-scribed. The negative control of cytotoxicity was selected using MG63 in complete medium without any treatment. Furthermore, the dose at which 50% of MG63 growth viability is inhibited (IC50) was calculated by means of nonlinear regression curve fit using GraphPad Prism 7.03.

### 2.5. Nasal Disinfectant Spray Performance

#### 2.5.1. Spray Coverage

Spray coverage of the NSD was evaluated through surface coverage and droplet analysis [39]. A blue dye [0.5% (*w*/*v*)] was added to the nasal disinfectant solution, which was then transferred into 30 mL amber commercial spray bottles. The spray heads were actuated three times onto white A4 paper placed vertically at a distance of 15 cm. After drying, the sheets were scanned at 600 dpi resolution. The images were processed using ImageJ 1.54p software (U.S. National Institutes of Health, Bethesda, MD, USA) applying a 4000 × 4000 pxl square crop, followed by thresholding. Spray coverage was quantified as the percentage of the outlined area converted to black, representing the true surface area covered.

#### 2.5.2. Storage Stability

We stored 30 mL of NSD in commercial amber glass containers at room temperature to verify their shelf life under storage conditions. Zeta potential, particle distribution, and absorbance measurements were taken three different times: after fresh synthesis, 6 months, and 1 year, to demonstrate the stability of the colloidal solution over time.

### 2.6. Molecular Docking Study

#### 2.6.1. Ligand Preparation

The ligand used for the molecular docking study was developed based on previous work by Ranoszek-Soliwoda et al. [40]. The 3D structure of the complex was downloaded from PubChem in SDF format. The molecule was optimized using the Avogadro program, choosing the Auto Optimization Tool, the MMFF94s force field, four steps per update, and the Gradient Descent algorithm. The optimized structure was saved in mol2 format.

The ligands were prepared in the AutoDockTools 1.5.6 software, following the protocol proposed by Rizvi et al. [41]. Briefly, the Gasteiger charges were calculated, the root in the torsion tree was selected and detected, and the torsions were defined. Finally, the ligands were saved in pdbqt format.

#### 2.6.2. Receptor Preparation

In order to evaluate the inhibitory potential of TA, we used the strategy of analyzing the interaction regions between the evaluated viral spikes and the angiotensin-converting enzyme 2 (ACE2). In the case of SARS-CoV-2, the viral spike is divided into seven regions, of which the RBD region is the first to establish contact with ACE2. These regions were identified as our active sites; thus, we downloaded the corresponding crystals of the most important variants of SARS-CoV-S identified at the peak time of the pandemic: Wild, Delta, and Omicron. The crystals were obtained from the RCSB Protein Data Bank (RCSB PDB) website. Table 1 describes each crystal for individual variants. For the evaluated models, the selection criteria were carried out following Holt et al., where crystals with a definition of less than 3.0 Å are prioritized [42]. All crystals were saved in PDB format.

The preparation of macromolecules was carried out in AutoDockTools 1.5.6 software following the protocol proposed by Rizvi et al. [41]. Briefly, Kollman charges representing the charge distribution of complex molecules and hydrogens were added, and water molecules were removed and saved in pdbqt format.

#### 2.6.3. Interactions and Binding Energy

The molecular docking results were evaluated in AutoDockTools. The selected active site corresponds to the location where the viral spike and ACE2 bind; therefore, the grid box was in the 94X, 126Y, 70Z coordinates, 0.5305 spacing, and a box size of −33.509X, 23.349Y, −2.466Z. Finally, for docking, Lamarckian GA (4.2) was performed.

The binding free energy was calculated by performing each study in triplicate in the AutoDockTools program. The interaction with the lowest free energy was selected from 10 docking conformations. The LigPlot+, Chimera, and Protein Ligand Interaction Profiler were used to elucidate the 2D and 3D interactions, and residues.

## 3. Results

The presence of AgNPs was analyzed by UV-visible spectroscopy, as shown in Figure 2a. The AgNPs exhibited a Ag surface plasmon resonance (SPR) with a classical wavelength of 404 nm [43,44]. The spectrum also displayed two peaks, at 250 nm and 278 nm, corresponding to π→π* transitions from SC and TA, respectively. Moreover, the incorporation of T80 did not alter the SPR of AgNPs in NSD. Interestingly, the particle size distribution, polarity, and zeta potential of the colloidal solutions are crucial factors in determining nanostructured uptake systems behavior. Therefore, we applied DLS, illustrating the particle size distribution of the AgNPs, which were monodispersed below 5 nm, illustrating a notably uniform dispersion, which indicated the high quality of the colloidal solution. Additionally, the AgNPs demonstrated high stability, as indicated by the negative zeta potential value of −44.77 ± 6.32 mV (Figure 2c). On the other hand, the addition of T80 increased the stability of the AgNPs showing a zeta potential of −52.77 ± 18.32 mV, showing that the T80 formulation did not alter the original charge due to its non-ionic nature.

Multiple mechanisms have been proposed for the synthesis of AgNPs using tannic acid (TA) as a reducing agent. The unique characteristics of tannic acid, a polyphenol-rich in –OH groups from catechin and gallic acid subunits, are highlighted. During the nucleation step, Ag⁺ ions are reduced by the electron-donating –OH groups from catechin and gallic acid [45,46]. This electron donation also generates catechin and gallic acid radicals, which continue to participate in the reduction of Ag^+^ to Ag^0^, leading to the formation of silver clusters, such as Ag_2_ and Ag_4_. These clusters promote the growth phase of AgNPs.

In parallel, sodium citrate plays a multifaceted role in the process. First, it acts as a stabilizer by coordinating with the surface of AgNPs, preventing aggregation [44]. Second, sodium citrate can participate in the reduction process; however, its reducing capacity is weaker than that of tannic acid [47]. Finally, the citrate ion imparts a negative charge to the surface of the AgNPs, promoting electrostatic repulsion and maintaining a stable colloidal system [48].

In a previous study of AgNP complex coordination and stabilization, Ranoszek-Soliwoda et al. [40] demonstrated that the formation of a TA-SC complex can act as a stabilizer for AgNPs. This mounting evidence was confirmed by the presence of the TA-SC complex in our AgNPs, determined using FTIR analysis, for which the results are shown in Figure 2d and Table 2. The TA-SC coordination complex exhibited the characteristic absorption bands at 3364, 2924, 2854, and 1576 cm^−1^, corresponding to the stretching vibrations of O-H, C-H, C=O, and C-O bonds, respectively [40,49]. In the fingerprint region, the TA-SC spectrum displayed absorption bands at 1392, 1264, and 1104 cm^−1^, attributed to the stretching vibrations of C-OH, C-H, and C-O-C, respectively [50]. A comparison with the FTIR spectrum of AgNPs shows that these functional group characteristic bands of the TA-SC persisted, confirming that the complex effectively caps the nanoparticles (in line with the zeta potential results). Moreover, it is important to highlight that the complex TSC-SC fingerprint is mainly detected in the experimental chelating models of AgNPs, and stabilizer for NSD formulation. This further supports the robust capability of the AgNPs to be incorporated into non-hydrophobic delivery systems. Therefore, we conclude that the AgNPs are enveloped by the TA-SC complex, contributing to the stability of the colloidal solutions.

SEM characterization revealed the morphology of the AgNPs and the AgNPs incorporated in NSD. Figure 3a,b show that the AgNPs exhibit a spherical shape with an average size of 11.10 ± 2.07 nm and 13.58 ± 2.92 nm for NSD. A slight discrepancy between the particle sizes measured by DLS and SEM is observed; however, this difference is expected due to the sample preparation and measurement conditions. It means an aqueous colloidal solution for DLS and dried for SEM. Additionally, EDS analysis confirms the formed spherical particles are indeed pure AgNPs.

TEM characterization revealed the morphology of the AgNPs and the AgNPs incorporated in NSD. Figure 4a,b show that the AgNPs exhibit a spherical shape with average sizes of 4.40 ± 0.44 nm and 5.04 ± 0.46 nm for NSD, respectively. In this case, the particle sizes measured by DLS and TEM are highly correlated, indicating consistency across the measurement techniques (Figure 4c). The SAED patterns further confirm the crystallinity of the AgNPs, revealing distinct diffraction rings corresponding to d-spacing of 0.24 nm and 0.23 nm, characteristic of the (111) crystallographic plane of FCC AgNPs (Figure 4d,e).

The MTT assay showed an IC50 value of 17.25 µg/mL for the AgNPs used in the NSD formulation, indicating that cell viability decreases only at higher concentrations. At lower concentrations (below IC50), the treated cells exhibited minimal morphological changes and maintained viability, as illustrated in Figure 5. These results suggest that the AgNP concentrations used are below the cytotoxic threshold, confirming their safety for human cells while preserving the antimicrobial efficacy of the formulation.

The atomizer spray pattern and coverage area are critical parameters for guaranteeing effective deposition of the atomized solution in the respiratory tract. Therefore, in this study, we compared the coverage performance of AgNPs and NSD (Figure 6a). Both solutions exhibited a complete cone spray pattern. The AgNPs covered a larger area (110 cm^2^) compared to the NSD (90 cm^2^), but with substantially increased sprayed droplets compared to the NSD. This is evident in the center of the spray images, where the AgNPs show runoff in the central zone (Figure 6b,c). A detailed analysis of different points within the coverage area further highlights the improved droplet spraying from the NSD. This behavior enhances adhesion, increases the surface area of the nasal mucosa covered by the solution, and helps control nasal drips, reducing the loss of the applied disinfectant and thereby improving protection against external infectious agents.

The stability and shelf life of a colloidal solution is critical when considering its use in products such as nasal disinfectants; thus, the AgNPs and NSD were tested for up to one year. DLS and UV-Vis were performed on freshly prepared samples, as well as after six and twelve months of storage. As shown in Figure 7a–c, the colloidal solutions exhibited remarkable stability over the twelve-month aging period, showing a slightly increase from 3.13 ± 0.36 nm to 3.55 ± 0.31 nm for the AgNPs and from 3.02 ± 0.69 nm to 3.30 ± 0.21 nm for the NSD, highlighting that the solutions did not undergo significant agglomeration during prolonged storage. In contrast, the zeta potential of the colloidal solution decreased from −42.82 ± 1.72 to −37.63 ± 0.93 mV for AgNPs and −50.84 ± 1.83 mV to −33.50 ± 1.92 mV for NSD over twelve months. However, it is relevant to hallmark that the zeta potential remains extensively above 30 mV, overcoming the solution’s stability period beyond the recommended six months [51]. Finally, the UV-Vis results showed that the SPR wavelength and the absorbance remained unchanged for both solutions. Despite the similar behavior of AgNPs and NSD, indicating that the addition of T80 does not significantly alter the colloidal system as expected, it is important to note the difference in absorbance. The higher absorbance of NSD compared to AgNPs suggests a more homogeneous particle size distribution and minimized Ag oxidation, which can be attributed to the key role of T80 in preventing aggregation and preserving the optical properties of AgNPs during storage.

The molecular docking analysis uncovered promising insights into the interaction between the SARS-CoV-2 spike protein and the human ACE2 receptor. The binding energies of the TA-SC complex between various SARS-CoV-2 variants were consistently below −8.0 kcal/mol, indicating a strong affinity, suggesting that the interactions are thermodynamically favorable and spontaneous (Figure 8). Notably, the wild-type variant of SARS-CoV-2 exhibited the lowest binding energy, hinting at its potentially stronger interaction with the TA-SC complex than the other variants. These findings hold significant promise for the development of potential inhibitors, antagonist receptors, Ag-complex agonists (as observed here), and drug development.

Our docking analysis also revealed a crucial aspect of the interaction between the TA-SC complex and the spike protein variants. Figure 9, Figure 10, Figure 11 and Figure 12 represent the key amino acid residues and the specific chemical interactions observed in each variant.

## 4. Discussion

AgNPs stand as the most relevant and versatile nanostructured materials for medical applications, due to the provided long-lasting and effective broad-spectrum antimicrobial activity [52,53,54]. Furthermore, it is of great importance to consider the comprehensive physicochemical parameters that can modulate the AgNPs’ response to potentially prevent respiratory infections such as those related to SARS-CoV-2 via nasal application. Although, previous studies of antiviral metallic nanoparticles have suggested that the size (60–120 nm or 80–150 nm), the structural composition (e.g., enveloped virus), and the molecular topography (receptors) of SARS-CoV-2 viral particles can modulate the viral–surface binding response, contributing to viral eradication [55,56]. Therefore, it is feasible to engineer novel nanodisinfecting formulations that can selectively respond to potential viral molecules transmitted by non-invasive administration routes.

Previous reports of broad-spectrum antimicrobial activity have shown that small distributed AgNPs (2–15 nm) can produce long-lasting SARS-CoV-2 inhibition and subsequent disruption of secondary related opportunistic infections [57,58]. Moreover, mounting evidence supports these results, illustrating robust viral inactivation against H5N1 by 5–13 nm AgNPs [59]. Interestingly, Valdez-Salas et al. [59] demonstrated that a AgNP-based disinfectant can disrupt enveloped influenza-coated viruses in a log time reduction of 5 min. Moreover, the potential effectiveness of these disinfectants extended to stiffer and thicker bacterial and fungal cellular structures, disrupting the microbial growth capability of the AgNP penetration to the sub-cellular level, which produced bacterial/fungal disruption and prominently cell death. Thus, the aforementioned evidence illustrated that our nanoparticles are exquisitely engineered in a 3 to 6 nm size distribution, which is appropriate for the suppression of SARS-CoV-2 and more microorganisms involved in respiratory diseases.

It is of further importance to consider the zeta potential in colloidal solutions, which defines the nanoparticle charge, and far more attractive, the colloidal solution stability. Thus, when considering the antimicrobial applications orchestrated by the zeta potential behavior, several studies have suggested that the surface charge can contradictorily influence both the microbial binding and disrupting capability of AgNPs. Interestingly, Abbaszadegar et al. and Ahmad et al. [60,61] concluded that positively charged AgNPs have a superior antimicrobial activity compared to those negatively charged due to associated electrostatic attractions. On the other hand, Ferreyra-Maillard et al. and Salvoni et al. [62,63] suggested that AgNPs showing a negative zeta potential not only allowed for potent antimicrobial activity but also low cytotoxicity, resulting from a non-disrupting interaction between the polar heads of membrane lipids through electrostatic attraction. Therefore, nanoparticles with negative zeta potential may be ideal for cosmetic preservatives and pharmaceutical preparations that can be applied to humans and animals by different administration routes. Therefore, the latter was a hallmarking desired route of administration in our colloidal solution, since both AgNPs and NSD had a robust negative zeta potential endowing high stability.

It is noteworthy to mention that the nanostructured morphology can orchestrate a regulating factor influencing antimicrobial activity fine-tuned by larger area surfaces, presenting greater toxicity against viruses, bacteria, and fungi [64,65,66]. In other words, this information collectively means that spherical nanoparticles can represent the optimal morphology for medical applications [67]. In order to derive the maximum surface area, our synthesis was focused on obtaining uniform and small spherical AgNPs.

On the other hand, our molecular docking study demonstrated that the TA-SC complex of the capping agent interacts with pivotal residues of the spike and ACE2-binding protein interface (Appendix A), potentially impacting the mechanism of viral replication [68,69,70,71]. Forefront examples of these residues include Lys417, Tyr449, Gln493, Gln498, and Asn501, which stabilize the interaction between the virus and ACE2, facilitating viral entry into human cells. These residues also contribute to increased resistance to neutralizing antibodies and evasion of immune detection.

When comparing our binding energy results with FDA-approved antibodies for COVID-19 prophylaxis (tixagevimab, pemivibart, and cilgavimab), only one study has reported the molecular docking and binding energies of tixagevimab and cilgavimab on the receptor-binding domain (RBD) of the SARS-CoV-2 spike protein for the Delta variant [72,73]. The reported values in this study were −13.9 kcal/mol for tixagevimab and −12.4 kcal/mol for cilgavimab. In contrast, the binding energy of the TA-SC complex in our study for the Delta variant was −8.22 kcal/mol. While this difference is notable, it suggests that the TA-SC complex could still compete in binding affinity. Interestingly, we obtained stronger binding energies for the wild-type (−10.05 kcal/mol) and Omicron (−9.62 kcal/mol) variants, indicating that the TA-SC complex presents a more promising prophylactic potential against these variants than Delta.

Different phenolic compounds, including caffeic acid, chlorogenic acid, gallic acid, curcumin, and TA, have been shown to suppress SARS-CoV-2 by targeting different viral binding regions [74,75,76,77,78]. Elfiky [76] determined by molecular docking and bioinformatics tools that caffeic acid and cis-p coumaric acid are potential inhibitors of SARS-CoV-2 by interference between the spike protein (C480-488 region) and the SARS-CoV-2 receptor host cell (GRP78). These phenolic compounds act at the binding sites through hydrogen bonding and robust hydrophobic interactions, resulting in an energy of −5.2 kcal/mol for caffeic acid and −5.6 kcal/mol for p-cis coumaric acid. These cumulative end results are important since GRP78 is a protein encoded in the HSPA5 gene and is generally expressed in various tissues, including bronchial epithelial cells and respiratory mucosa. Moreover, Wang et al. [78] showed that TA inhibited more than 90% of the enzymatic activity of SARS-CoV-2 Mpro at a concentration of 50 µM and a mean maximum inhibitory concentration of 0.13 µM, as observed from a dose–response analysis. Collectively, the molecular docking suppression mechanism suggested that TA occupies a defined space by binding the main *Cys145* residue via pi-sulfur and hydrogen bonding. Similarly, *Asn142* and *Met165* were extensively coupled by hydrogen bonds and pi-alkyl interactions. Using the same enzyme, Coehlo et al. [75] obtained a mean maximum inhibitory concentration of 2.1 µM, thus far supporting our breakthrough results of SARS-CoV-2 binding interactions with TA-SC complexing AgNPs.

Although most of the evidence regarding phenolic compounds has been proven by in silico studies, Goc et al. [77] obtained in vitro experimental evidence for 56 phenolic compounds, including TA, studying their activity against TMPRSS2, Cathepsin L, and ACE2. The results showed that the phenolic acids that bound to the greatest extent with the receptor-binding domain of SARS-CoV-2 (in a A549 cells) were curcumin (100.0 ± 2) followed by TA (79.4 ± 2.3). This groundbreaking information is in line with our current results of molecular docking interactions for capped AgNPs in a TA-SC complex and the different variants of SARS-CoV-2.

Far more critical are our results, which agree with the published literature, showing that our synthesis method is in the range of these active concentration responses. Therefore, our experimental AgNP doses can act in two ways for eradicating respiratory infections. The first is through obtaining nanoparticles smaller than 20 nm and the second with a concentration that has turned out to inhibit the enzymatic activity of SARS-CoV-2 without disrupting the epithelial mucous cellular viability [79].

The structure of the nasal cavity is composed of the vestibule, atrium, respiratory region, olfactory region, and nasopharynx, characterizing the respiratory region as the largest absorption area. Among these elements that constitute the respiratory region are the basal, calceiform, ciliary, and mucosal cells [80]. In particular, these regions are of utmost importance because the ACE2 and TMPRSS2 receptors are expressed and co-expressed in them, which are cataloged as the gateway for SARS-CoV-2 cellular binding [81]. The first defense that these cells have is the mucosa, a conglomerate of hydrophilic polysaccharides and permeable high-density polymers. Therefore, the affinity of the formulation to create a barrier in the nasal mucosa is one of the most relevant strategies to prevent contagion upon the arrival of SARS-CoV-2 at the ACE2 and TMPRSS2 receptors [82]. The results of spray coverage in our formulation, with or without surfactant, proved to have a wide coverage, which can create an antimicrobial barrier for the nasal mucosa. This prevents a significant amount of the formulation from being dispelled through the nasal cavities and conserves a superior coverage area with the addition of the surfactant (T80). Similarly, the formulation with surfactant caused smaller droplets with a more uniform distribution. This improvement in the spraying of the formulation is supported by the behavior of the surfactants in reducing surface tension. This amplifies the dispersion of our NSD by T80 due to the reduction in surface tension and the extensive affinity with the nasal mucosa brought by improving the wettability behavior of the epithelial–NSD surfaces. Furthermore, the physical chemical characteristics of T80 have been shown to have special attributes for nasal and cerebral drug release, such as biocompatibility, drug delivery, mucoadhesion, and permeability along the nasal mucosa [83,84,85].

Other proposals for spray nasal disinfectants emphasize the coverage area, seeking to increase it by adding polysaccharides in the formulation. For instance, Moakes et al. [39] formulated a spray disinfectant to prevent COVID-19, where the active ingredient was Iota-carrageenan. Furthermore, the addition of gellan gum (another polysaccharide) showed an increased coverage area for all the tested ratios when considering a 5% coverage using iota-carrageenan and 25% coverage for a 50:50 gellan gum:Iota–carrageenan ratio. Using the same active ingredients, Robinson et al. [86] improved the coverage and mucoadhesion of their formulation with the addition of low-acyl gellan gum. The largest coverage area was obtained with a combination of 0.25% Iota-carrageenan and 0.25% low-acyl gellan gum (%*w*/*v*), illustrating a 20 cm^2^ coated area. It is important to highlight that, for our work, the coverage protocols were similar to those of Robinson et al., resulting in higher coverage for our NSD formulations (115 cm^2^) and without T80 (85 cm^2^). These results lead us to propose the exploration and use of several non-ionic surfactants since polysaccharide additives have generally been used to improve absorption to the nasal mucosa due to the antimicrobial activity that some may present. Nonetheless, polysaccharides could act as a detrimental platform for promoting several microbial growth activities, thus negatively affecting the antimicrobial requirements of the active delivery components.

Finally, part of the atomized amount does not remain impregnated in the mucosal wall, which may represent an advantage for nasal formulations. Due to the size of the nanoparticles in the formulation, they can move to the lower respiratory tracts or even the alveoli by means of Brownian movement, with improved transmission thanks to mechanical parameters such as the diffusion capability of the disinfectant. The ability of nanoparticles to reach the alveoli is intended to target advanced stages of respiratory infections. For instance, in SARS-CoV-2, studies have shown that the most severe phase of COVID-19 occurs when the alveolar sacs fill with fluid, obstructing gas exchange and leading to systemic damage, including to the nervous and immune systems. Researchers should take forward this collective information and the results of the NSD, to increase the coverage of attraction to the mucosal surface area, via the potential affinity of the AgNPs complexed in the TA-SC with SARS-CoV-2. The present work can open up a potential role in developing advanced antiviral disinfectants as an effective protection barrier, providing a long-lasting, accessible administration route available for seasonal and potentially pandemic diseases. However, more studies are recommended to support the present results.

## 5. Conclusions

Nasal sprays offer a promising alternative to conventional drug administration routes such as oral, intramuscular, and transcutaneous methods, particularly for assessing the respiratory pathway. However, it is crucial to consider key physicochemical factors that facilitate crossing this biological barrier. In this study, we developed an NSD colloidal based-solution that fulfilled the synthesis of spherical AgNPs with a uniform size distribution (<20 nm), a negative surface charge, and high stability (<−30 mV). The enhancement of stability and mucoadhesion was achieved with the formation of a tannic acid–sodium citrate complexing with the AgNPs and the addition of T80, a non-ionic surfactant.

Moreover, molecular docking results evidenced prominent binding interactions between TA-SC-stabilized AgNPs and key receptors in ACE2 and the SARS-CoV-2 spike protein from the main variants (Wuhan, Delta, and Omicron). These findings suggest that NSD is not only suitable for nasal administration but also has significant potential in providing efficient coverage and sustained delivery to the respiratory tract.

Further research, including preclinical and clinical tests, is necessary to validate the safety, efficacy, and antiviral properties of our formulation. Altogether, the resulting NSD demonstrated the desired characteristics for nasal administration and efficient coverage upon atomization. Far more important, our findings indicate that this NSD shows promise as a potential nasal spray formulation.

## Figures and Tables

**Figure 1 nanomaterials-15-00533-f001:**
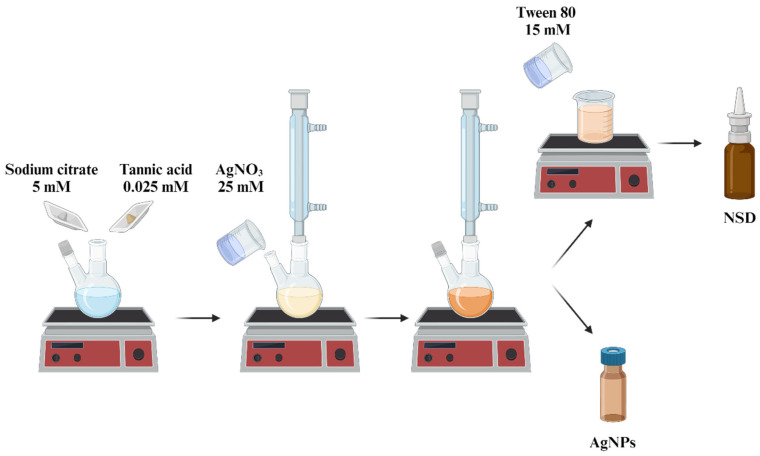
Schematic illustration of the synthesis of AgNPs through the bottom-up route, followed by the preparation of the NSD. Illustration highlights the key reagents and final formulation process.

**Figure 2 nanomaterials-15-00533-f002:**
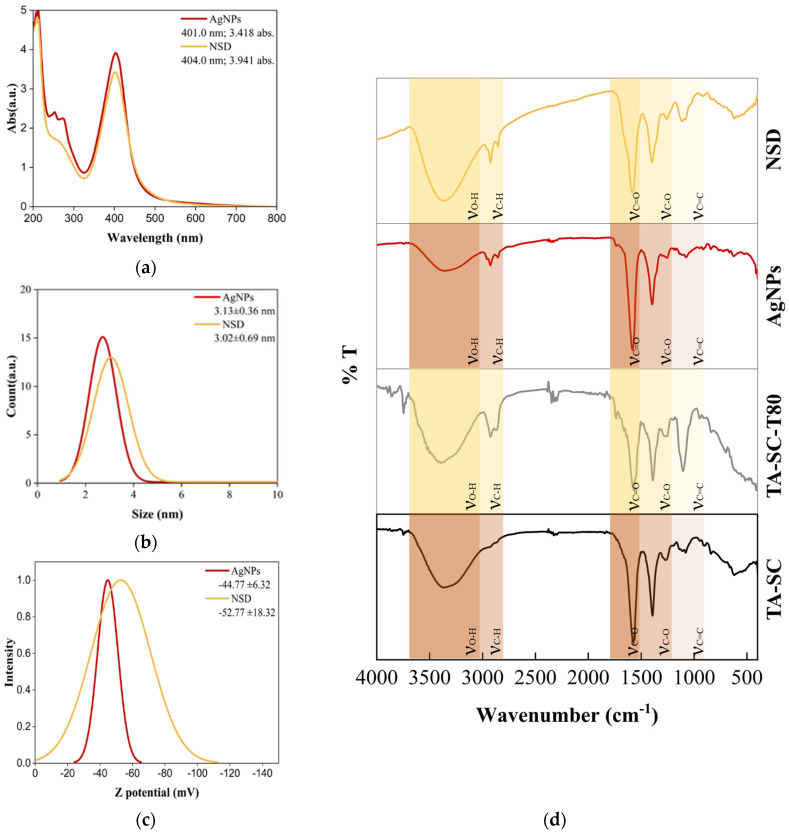
Physicochemical characterization of AgNPs and NSD: (**a**) SPR showing characteristic absorption peak; (**b**) size distribution; (**c**) zeta potential, and (**d**) FTIR spectra highlighting functional groups.

**Figure 3 nanomaterials-15-00533-f003:**
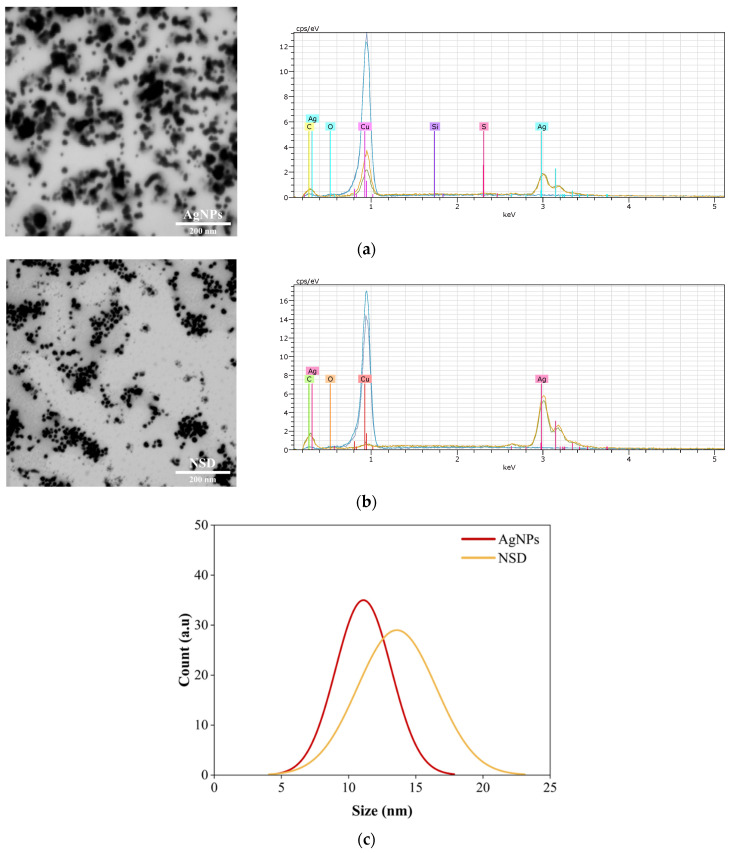
SEM micrographs and EDX spectra of (**a**) AgNPs; (**b**) NSD; (**c**) particle size distribution from SEM micrographs. All micrographs were captured at 25,000× magnification.

**Figure 4 nanomaterials-15-00533-f004:**
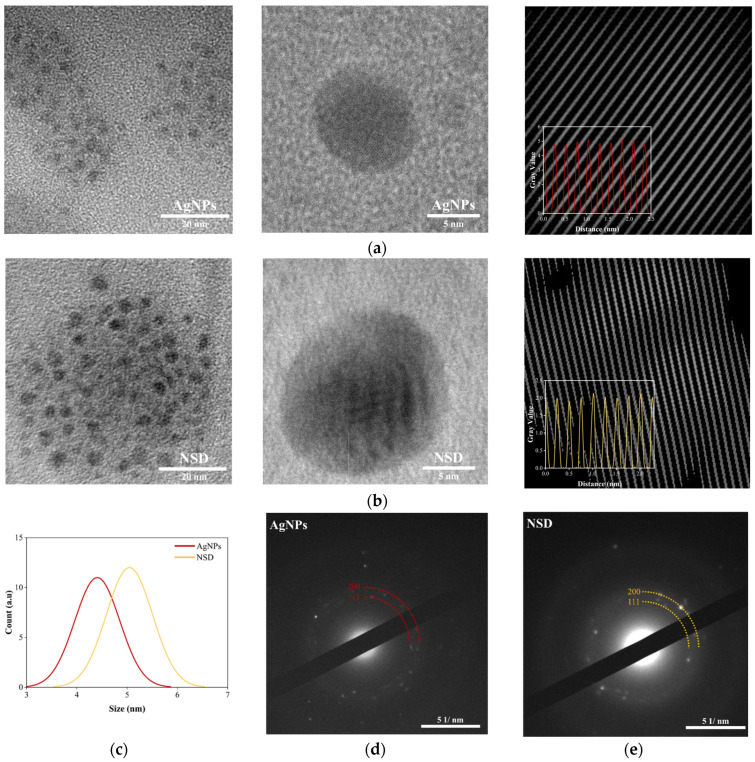
TEM, HR-TEM, and d-spacing analysis of (**a**) AgNPs and (**b**) NSD; (**c**) particle size distribution from TEM images; SAED pattern of (**d**) AgNPs and (**e**) NSD. HR-TEM images were captured at 500,000× magnification.

**Figure 5 nanomaterials-15-00533-f005:**
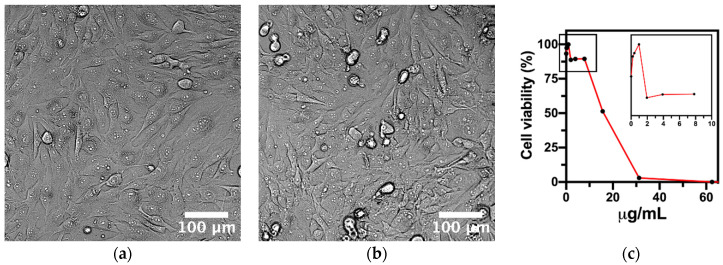
Micrographs showing the morphology of MG63: (**a**) untreated control group, (**b**) cells treated with AgNPs, and (**c**) cell viability (%) as determined by the MTT assay at different AgNP concentrations (0–60 μg/mL). Scale bar = 100 µm.

**Figure 6 nanomaterials-15-00533-f006:**
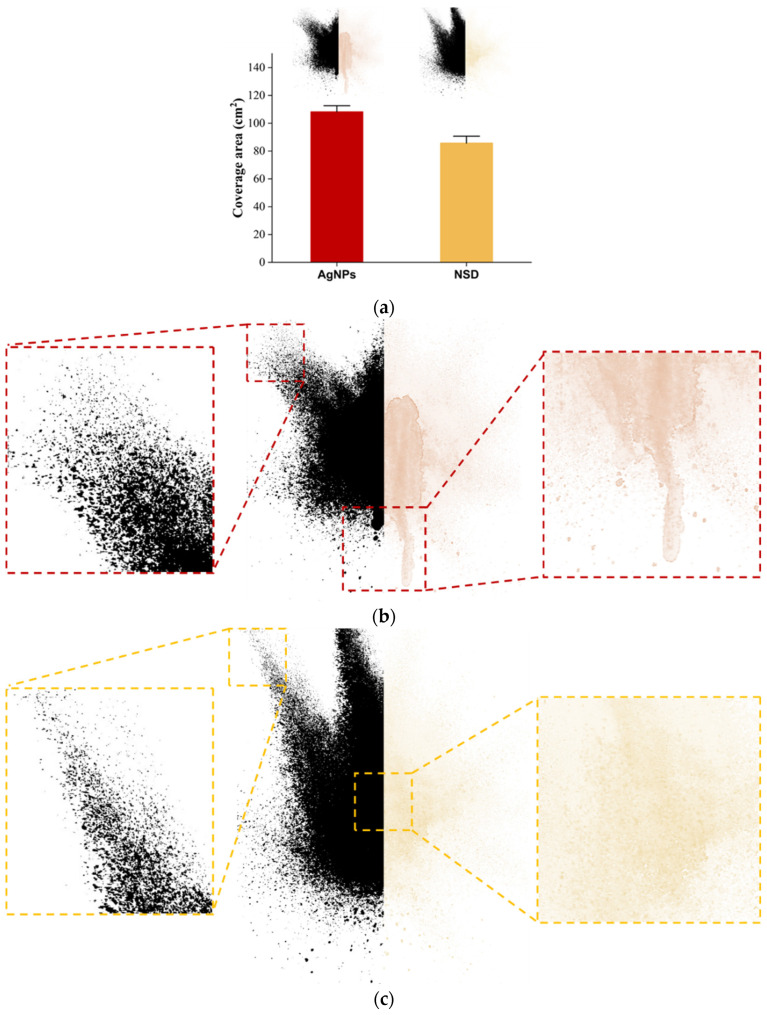
Spray ability performance of AgNPs and NSD: (**a**) coverage area comparison; detailed analysis of spray pattern and drop formation of (**b**) AgNPs and (**c**) NSD.

**Figure 7 nanomaterials-15-00533-f007:**
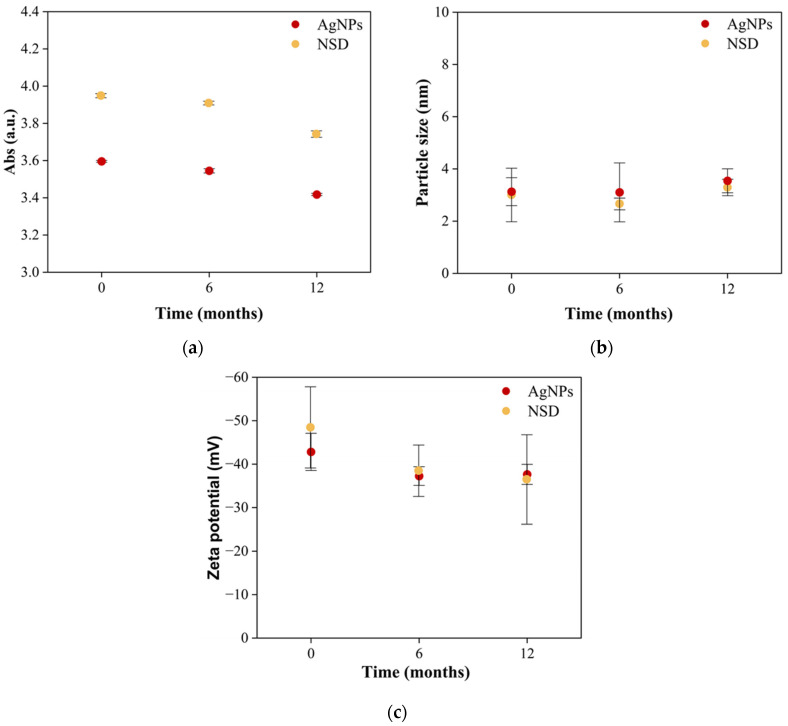
Storage stability behavior of AgNPs and NSD: (**a**) SPR analysis; (**b**) particle size; evolution and (**c**) zeta potential.

**Figure 8 nanomaterials-15-00533-f008:**
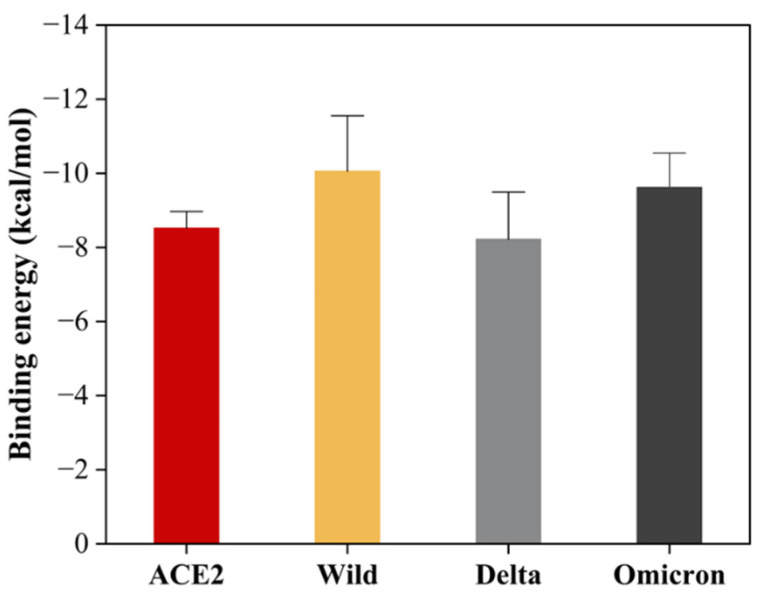
Binding energy of TA-SC complex against ACE2 and SARS-CoV-2 variants.

**Figure 9 nanomaterials-15-00533-f009:**
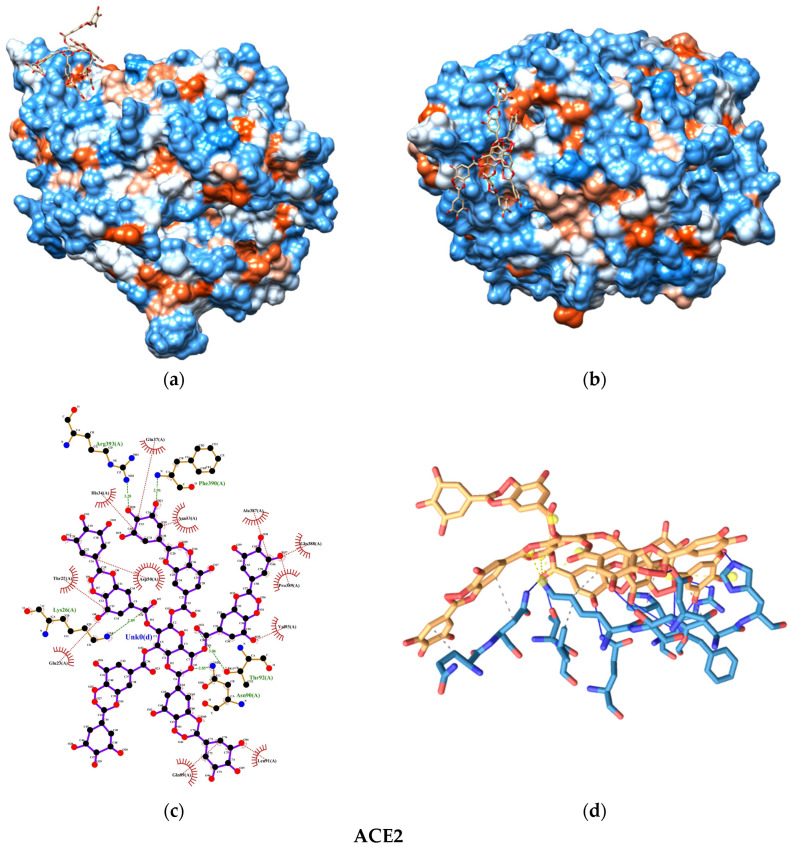
Molecular docking test between TA-SC complex and ACE2: (**a**) 3D side view; (**b**) 3D top view; (**c**) 2D main interactions; and (**d**) 3d main interactions. Amino acids color-coded (red hydrophobic; blue hydrophilic).

**Figure 10 nanomaterials-15-00533-f010:**
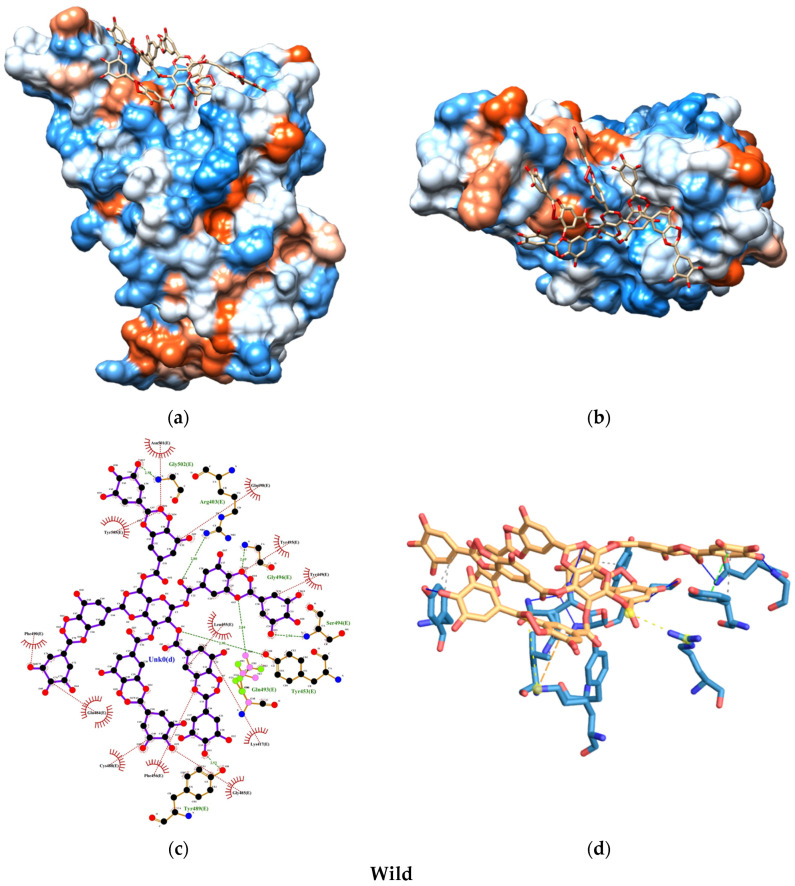
Molecular docking test between TA-SC complex and wild variant: (**a**) 3D side view; (**b**) 3D top view; (**c**) 2D main interactions; and (**d**) 3d main interactions. Amino acids color-coded (red hydrophobic; blue hydrophilic).

**Figure 11 nanomaterials-15-00533-f011:**
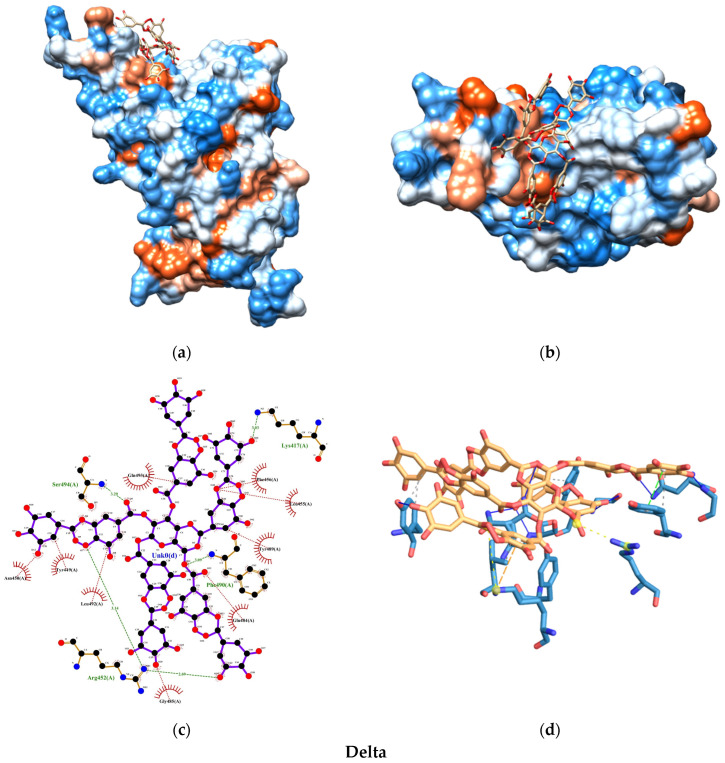
Molecular docking test between TA-SC complex and Delta variant: (**a**) 3D side view; (**b**) 3D top view; (**c**) 2D main interactions; and (**d**) 3D main interactions. Amino acids color-coded (red hydrophobic; blue hydrophilic).

**Figure 12 nanomaterials-15-00533-f012:**
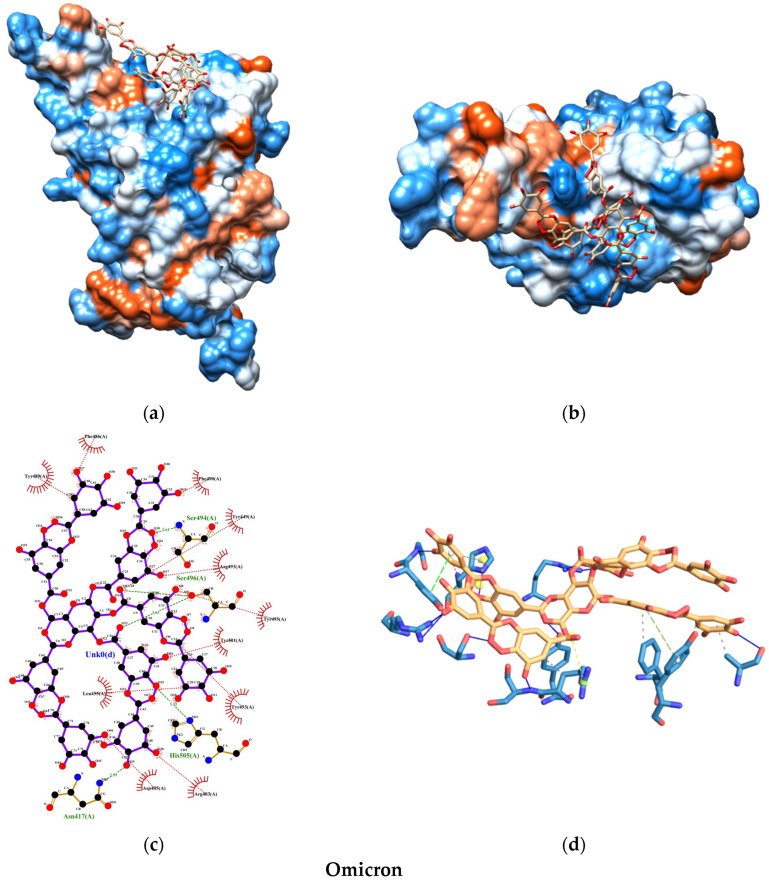
Molecular docking test between TA-SC complex and Omicron variant: (**a**) 3D side view; (**b**) 3D top view; (**c**) 2D main interactions; and (**d**) 3D main interactions. Amino acids color-coded (red hydrophobic; blue hydrophilic).

**Table 1 nanomaterials-15-00533-t001:** Information of crystals used for molecular docking study.

Crystal	Description	Method	Resolution (Å)
6MOJ	ACE2	X-ray diffraction	2.43
6MOJ	Wild	X-ray diffraction	2.43
7V8B	Delta	Cryo-electron microscopy	3.20
7T9L	Omicron	Cryo-electron microscopy	2.66

**Table 2 nanomaterials-15-00533-t002:** Wavenumbers (in cm^−1^) of selected vibrations of complex, AgNPs, and NSD, corresponding to the peaks marked in Figure 2d.

TA-SC	TA-SA-T80	AgNPs	NSD	Description
3364	3364	3364	3364	ν(O-H)
2924	2924	2924	2924	ν(C-H)
2854	2854	2854	2854	ν(C=O)
-	1736	1734	-	ν(C=O)
1576	1572	1586	1586	ν(C=O)
1392	1390	1398	1398	ν(C-OH)
1264	1262	1260	1260	ν(C-H)
-	1104	-	1114	ν(C-O-C)

## Data Availability

The original contributions presented in this study are included in the article/Appendix A. Further inquiries can be directed to the corresponding author.

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
