# Peer review of "Nasal Spray Disinfectant for Respiratory Infections Based on Functionalized Silver Nanoparticles: A Physicochemical and Docking Approach"

_nanomaterials, 2025, doi:10.3390/nano15070533_

Round 1
Reviewer 1 Report
Comments and Suggestions for Authors
The manuscript presents a study on functionalized silver nanoparticles (AgNPs) designed for use in nasal spray disinfectants. While the study includes physicochemical characterizations, stability tests, and molecular docking analyses to evaluate the efficacy of the AgNPs, the manuscript suffers from significant methodological, analytical, and presentation issues that severely undermine its scientific rigor. Given the extensive deficiencies outlined above, including the lack of essential characterizations (XRD, HRTEM, XPS), incomplete discussion of stability and synthesis mechanisms, and absence of biological validation, the manuscript does not meet the necessary standards for publication in a specialized nanomaterials journal. I recommend rejection of the manuscript. If the authors wish to resubmit to a different venue or this journal, they must comprehensively address all the aforementioned issues, perform additional critical experiments, and significantly improve data interpretation and presentation. Below are the critical concerns:
#1 The study lacks a thorough structural and crystallographic characterization of the AgNPs. XRD and HRTE are essential for confirming crystallinity, phase purity, and morphology. Without XRD, the crystalline nature of the AgNPs remains uncertain, and UV-Vis spectroscopy alone is inadequate to determine structural attributes or verify the formation of pure, highly crystalline nanoparticles. HRTEM is crucial for resolving lattice fringes, identifying core-shell structures, and assessing nanoparticle dispersity at the atomic level. The omission of these critical analyses significantly undermines the completeness and reliability of the nanoparticle characterization.
#2 The manuscript reports particle size data from SEM and DLS, which are not sufficient for a precise determination of nanoparticle size. SEM is not an ideal technique for measuring nanoparticle dimensions in solution, as it only provides surface morphology and cannot distinguish between primary nanoparticles and aggregates. HRTEM should be employed for accurate particle size measurements. Furthermore, DLS should be repeated over time to assess long-term colloidal stability.
#3 The manuscript relies on EDX for elemental analysis; however, EDX cannot determine oxidation states or phase purity. Complementary XPS is required to confirm the oxidation state of silver, particularly since AgNPs are prone to oxidation into Ag₂O, which can influence their physicochemical properties and stability.
#4 The UV-Vis results show a SPR peak at 404 nm, but no comparison is made with reference spectra for AgNPs of different sizes. The manuscript should analyze the UV-Vis spectrum over time to monitor peak shifts that indicate aggregation or stability concerns. Additionally, a quantitative discussion on how the observed spectra correlate with particle size is necessary.
#5 The authors report a negative zeta potential as evidence of good colloidal stability but fail to conduct long-term stability testing. Repeated DLS measurements at different time intervals are required to confirm stability over extended periods.
#6 The discussion of FTIR spectra lacks depth. A more detailed comparison with reference spectra is needed to confirm the functionalization of AgNPs with tannic acid and sodium citrate. The role of these stabilizers in nanoparticle formation and interaction should be elucidated.
#7 The authors fail to provide a reaction mechanism describing the reduction of Ag⁺ to Ag⁰, nucleation, and growth stages. A well-defined mechanistic explanation is crucial, especially for high-impact journals in nanomaterials science. Additionally, the presence of Ag₂O as a byproduct should be investigated, as it may influence the optical and functional properties of the AgNPs.
#8 The molecular docking study suggests potential antiviral interactions but does not substitute for biological validation. The limitations of in silico studies should be acknowledged, and experimental antiviral assays are required to substantiate the theoretical claims.
#9 Before proposing nasal spray applications, cytotoxicity and biocompatibility tests must be conducted to ensure the AgNPs are safe for human use. Tests such as MTT assays on nasal epithelial cells would significantly strengthen the manuscript’s claims.
#10 The manuscript lacks quantification of spray retention and mucoadhesion, which are critical for evaluating nasal spray efficacy. Additional experiments using fluorescence labeling or rheological analysis should be performed.
#11 The manuscript discusses AgNPs’ potential for viral inhibition but does not compare its findings with previously published antiviral AgNP formulations. A comparison with state-of-the-art literature is necessary to establish the novelty and relevance of the work.
#12 AgNPs are known to aggregate under certain conditions, yet the manuscript does not discuss potential aggregation mechanisms or mitigation strategies. Without this information, the stability claims remain unsubstantiated.
#13 The total silver content in the formulation should be quantified using Inductively Coupled Plasma Mass Spectrometry (ICP-MS) or Inductively Coupled Plasma Optical Emission Spectrometry (ICP-OES) to ensure consistency in the formulation.
#14 The manuscript inconsistently uses the terms "AgNPs," "colloidal silver," and "silver nanoparticles." A standardized nomenclature should be adopted for clarity.
#15 Given the potential environmental risks of AgNPs, the manuscript should briefly discuss toxicity and biodegradability concerns, particularly regarding environmental accumulation.
#16 The conclusion does not outline future research directions, particularly regarding experimental validation of antiviral activity and in vivo safety studies. A discussion on the next steps would enhance the manuscript’s impact.
#17 The similarity score of 28% is relatively high and raises concerns about possible text overlap with existing literature. The authors must reduce this similarity score to avoid potential ethical concerns during peer review.
Author Response
The manuscript presents a study on functionalized silver nanoparticles (AgNPs) designed for use in nasal spray disinfectants. While the study includes physicochemical characterizations, stability tests, and molecular docking analyses to evaluate the efficacy of the AgNPs, the manuscript suffers from significant methodological, analytical, and presentation issues that severely undermine its scientific rigor. Given the extensive deficiencies outlined above, including the lack of essential characterizations (XRD, HRTEM, XPS), incomplete discussion of stability and synthesis mechanisms, and absence of biological validation, the manuscript does not meet the necessary standards for publication in a specialized nanomaterials journal. I recommend rejection of the manuscript. If the authors wish to resubmit to a different venue or this journal, they must comprehensively address all the aforementioned issues, perform additional critical experiments, and significantly improve data interpretation and presentation. Below are the critical concerns:
- The study lacks a thorough structural and crystallographic characterization of the AgNPs. XRD and HRTE are essential for confirming crystallinity, phase purity, and morphology. Without XRD, the crystalline nature of the AgNPs remains uncertain, and UV-Vis spectroscopy alone is inadequate to determine structural attributes or verify the formation of pure, highly crystalline nanoparticles. HRTEM is crucial for resolving lattice fringes, identifying core-shell structures, and assessing nanoparticle dispersity at the atomic level. The omission of these critical analyses significantly undermines the completeness and reliability of the nanoparticle characterization.
Response 1: Thank you for your recommendation. We included HRTEM and SAED of AgNPs in the revised manuscript. With this characterization results, we confirmed the FCC crystalline structure of AgNPs.
L130-140: 3.3.4. Transmission Electron Microscopy (TEM) and Selected Area Electron Diffraction (SAED)
The morphology and size of the AgNPs were characterized through transmission electronic microscopy (TEM) by using a JEOL 2010 microscope operating at an accelerating voltage of 200 kV. The samples were prepared by depositing a 10 µL of AgNPs colloid and NSD into a carbon-coated Cu cell, followed by a drying process at room temperature in a desiccator. The crystalline structure of AgNPs was studied through selected area electron diffraction (SAED), obtained during the TEM characterization. The crystallographic phases were identified by comparing diffraction patterns obtained by the information provided by the Joint Committee on Powder Diffraction Standards (JCPDS), using JCPDS 04–0783 crystallographic chart.
L286-294: TEM characterization revealed the morphology of the AgNPs and the AgNPs in-corporated in NSD. Figures 4a and 4b show that the AgNPs exhibit a spherical shape with an average size of 4.40 ± 0.44 nm and 5.04 ± 0.46 nm for NSD, respectively. In this case, the particle sizes measured by DLS and TEM are highly correlated, indicating consistency across the measurement techniques (Figure 4c). The SAED patterns further confirm the crystallinity of the AgNPs, revealing distinct diffraction rings corresponding to a d-spacing of 0.24 nm and 0.23 nm, characteristic of the (111) crystallographic plane of FCC AgNPs (Figure 4d and 4e).
- The manuscript reports particle size data from SEM and DLS, which are not sufficient for a precise determination of nanoparticle size. SEM is not an ideal technique for measuring nanoparticle dimensions in solution, as it only provides surface morphology and cannot distinguish between primary nanoparticles and aggregates. HRTEM should be employed for accurate particle size measurements. Furthermore, DLS should be repeated over time to assess long-term colloidal stability.
Response 2: We have support particle size by HRTEM. Addressing to your concern about long-term colloidal stability, we have confirmed by monitoring zeta potential, particle size and absorbance at 404 nm for 1-year period.
- The manuscript relies on EDX for elemental analysis; however, EDX cannot determine oxidation states or phase purity. Complementary XPS is required to confirm the oxidation state of silver, particularly since AgNPs are prone to oxidation into Ag₂O, which can influence their physicochemical properties and stability.
Response 3: We appreciate and agree your suggestion regarding the characterization by XPS to confirm the oxidation AgNPs. However, we confirmed the prevention of oxidation of Ag NPs by UV-Vis spectroscopy. Specially, the characteristic SPR peak of AgNPs at 404nm remains intact during our storage study (1 year). The oxidation of Ag is cleared presented by the extinction of characteristic SPR (404 nm) and the formation of new peaks along the wavelength.
- The UV-Vis results show a SPR peak at 404 nm, but no comparison is made with reference spectra for AgNPs of different sizes. The manuscript should analyze the UV-Vis spectrum over time to monitor peak shifts that indicate aggregation or stability concerns. Additionally, a quantitative discussion on how the observed spectra correlate with particle size is necessary.
Response 4: Thank you for your comment. We have now added two references (40 and 41), which support the presence of a SPR at 404 nm for spherical Ag NPs, consistent with the findings in our work. Furthermore, as shown in Figure 5a, we have monitored the SPR absorption over a period of 1 year, demonstrating the high stability of the nanoparticles, indicating the absence of aggregation.
- The authors report a negative zeta potential as evidence of good colloidal stability but fail to conduct long-term stability testing. Repeated DLS measurements at different time intervals are required to confirm stability over extended periods.
Response 5: We would like to clarify that colloidal stability is not solely determined by the polarity of AgNPs. In our study, we used z potential as a key indicator of stability. A z potential above ± 30 mV is generally accepted as evidence of good colloidal stability.
To address the long-term stability, we have conducted z potential measurements over 1-year period, as shown in Figure 5c. These measurements confirm the high stability of AgNPs and NSD.
- The discussion of FTIR spectra lacks depth. A more detailed comparison with reference spectra is needed to confirm the functionalization of AgNPs with tannic acid and sodium citrate. The role of these stabilizers in nanoparticle formation and interaction should be elucidated.
Response 6: Thank you for your feedback. We have included three new references that support our FTIR spectra analysis. These reference support the peaks in AgNPs and NSD. Additionally, we have expanded the discussion to highlight the role of these stabilizers in the reduction and stabilization of AgNPs.
Ranoszek-Soliwoda, K.; Tomaszewska, E.; Socha, E.; Krzyczmonik, P.; Ignaczak, A.; Orlowski, P.; Krzyzowska, M.; Celichowski, G.; Grobelny, J. The Role of Tannic Acid and Sodium Citrate in the Synthesis of Silver Nanoparticles. J Nanopart Res 2017, 19 (8), 273. https://doi.org/10.1007/s11051-017-3973-9.
Ricci, A.; Olejar, K. J.; Parpinello, G. P.; Kilmartin, P. A.; Versari, A. Application of Fourier Transform Infrared (FTIR) Spectroscopy in the Characterization of Tannins. Applied Spectroscopy Reviews 2015, 50 (5), 407–442. https://doi.org/10.1080/05704928.2014.1000461.
Husanu, E.; Chiappe, C.; Bernardini, A.; Cappello, V.; Gemmi, M. Synthesis of Colloidal Ag Nanoparticles with Citrate Based Ionic Liquids as Reducing and Capping Agents. Colloids and Surfaces A: Physicochemical and Engineering Aspects 2018, 538, 506–512. https://doi.org/10.1016/j.colsurfa.2017.11.033.
- The authors fail to provide a reaction mechanism describing the reduction of Ag⁺ to Ag⁰, nucleation, and growth stages. A well-defined mechanistic explanation is crucial, especially for high-impact journals in nanomaterials science. Additionally, the presence of Ag₂O as a byproduct should be investigated, as it may influence the optical and functional properties of the AgNPs.
Response 7: Thank you for your advice. In response to your comment, we have now included a detailed reaction mechanism describing the reduction of Ag+ to Ag0, as well as the nucleation and growth stages involved in the formation of AgNPs.
L244-256: Multiple mechanisms have been proposed for the synthesis of AgNPs using tannic acid (TA) as a reducing agent. The unique characteristics of tannic acid, a polyphenol-rich in –OH groups from catechin and gallic acid subunits, are highlighted. During the nucleation step, Ag⁺ ions are reduced by the electron-donating –OH groups from catechin and gallic acid [43,44]. This electron donation also generates catechin and gallic acid radicals, which continue to participate in the reduction of Ag⁺ to Ag⁰, leading to the formation of silver clusters, such as Ag₂ and Ag₄. These clusters promote the growth phase of AgNPs.
In parallel, sodium citrate plays a multifaceted role in the process. First, it acts as a stabilizer by coordinating with the surface of AgNPs, preventing aggregation [42]. Second, sodium citrate can participate in the reduction process; however, its reducing capacity is weaker than that of tannic acid [45]. Finally, the citrate ion imparts a negative charge to the surface of the AgNPs, promoting electrostatic repulsion and maintaining a stable colloidal system [46].
- The molecular docking study suggests potential antiviral interactions but does not substitute for biological validation. The limitations of in silico studies should be acknowledged, and experimental antiviral assays are required to substantiate the theoretical claims.
Response 8: We confirm that molecular docking studies cannot replace in vivo and in vitro validation. However, molecular docking is a critical part of the early stages of drug design and evaluation, providing valuable insights into potential interactions. In addition, we deeply discuss our results with other studies that performed biological experiments.
- Before proposing nasal spray applications, cytotoxicity and biocompatibility tests must be conducted to ensure the AgNPs are safe for human use. Tests such as MTT assays on nasal epithelial cells would significantly strengthen the manuscript’s claims.
Respone 9: We include cytotoxicity test and MTT in the revised manuscript.
3.4. Cell culture and cytotoxicity test
L147-172: To analyze the cytotoxicity effect of the AgNPs, we performed the MTT (3-(4,5-dimethylthiazol-2-yl)-2,5-diphenyl tetrazolium bromide) viability test, as de-scribed previously [37,38]. For the viability test we used human osteosarcoma derived osteoblast-like cell line (MG-63, ATCC CRL-1427), cultured at 1 104 cells/mL in com-plete medium constituted of Dulbecco’s modified Eagle medium (DMEM, Thermo Fisher Scientific, USA) supplemented with 10% heat-inactivated fetal bovine serum (FBS, Thermo Fisher Scientific, USA) and 100 units/mL of penicillin-streptomycin (Thermo Fisher Scientific, USA) in separate 96-well flat-bottom polystyrene culture plates (Corning, USA) at 37°C in a humidified 5% CO2 incubator for 24h. Then, the cells were washed thrice for 5 min with warm 1 × phosphate buffered saline (PBS) and incubated for 24h with serial dilutions (62.5 µg/mL - 0.4 µg/mL) of the experimental AgNPs suspended in complete medium. The AgNPs doses were discarded, and the cells were carefully washed thrice with warm PBS, to remove any artifact present in the cultures. Afterwards, the MTT protocol was carried out by adding 100 µL of MTT (5 mg/mL, Sigma Aldrich, USA) to each well of the cultured 96-well plate and incubated at 37°C in a humidified 5% CO2 incubator for 3h. The resulting formazan crystals were dissolved by removing the remaining medium containing MTT. The plate was placed in an orbital shaker at 140 rpm with 200 µL of dimethyl sulfoxide (Sigma Aldrich, USA) for 20 min. Next, the optical density (O.D.) of the dissolved crystals was meas-ured at 590 nm using a microplate reader (Thermoskan, Thermo Fisher Scientific, USA). The baseline controls were measured in a series of culture wells containing AgNPs at the experimental dilutions and prepared for MTT without MG63 as de-scribed. The negative control of cytotoxicity was selected using MG63 in complete me-dium without any treatment. Furthermore, the dose at which 50% of MG63 growth viability is inhibited (IC50) was calculated by means of a nonlinear regression curve fit using GraphPad Prism 7.03.
L300-306: The MTT assay showed an IC50 value of 17.25 µg/mL for the AgNPs used in the NSD formulation, indicating that cell viability decreases only at higher concentrations. At lower concentrations (below IC50), the treated cells exhibited minimal morpholog-ical changes and maintained viability, as illustrated in Figure 5. These results suggest that the AgNP concentrations used are below the cytotoxic threshold, confirming their safety for human cells while preserving the antimicrobial efficacy of the formulation.
- The manuscript lacks quantification of spray retention and mucoadhesion, which are critical for evaluating nasal spray efficacy. Additional experiments using fluorescence labeling or rheological analysis should be performed.
Response 10: Spray retention and mucoadhesion was supported by spray coverage test (Figure 4) following previous reported protocol. We now include this reference in materials and methods section.
Moakes, R. J. A.; Davies, S. P.; Stamataki, Z.; Grover, L. M. Formulation of a Composite Nasal Spray Enabling Enhanced Surface Coverage and Prophylaxis of SARS‐COV‐2. Advanced Materials 2021, 33 (26), 2008304. https://doi.org/10.1002/adma.202008304.
- The manuscript discusses AgNPs’ potential for viral inhibition but does not compare its findings with previously published antiviral AgNP formulations. A comparison with state-of-the-art literature is necessary to establish the novelty and relevance of the work.
Response 11: We appreciate your suggestion regarding a comparison with previously published antiviral formulations. In the manuscript, we have discussed the antiviral inhibition of AgNPs, particularly in lines 378-386.
- AgNPs are known to aggregate under certain conditions, yet the manuscript does not discuss potential aggregation mechanisms or mitigation strategies. Without this information, the stability claims remain unsubstantiated.
Response 12: We studied the potential aggregation and mitigation adding a non-ionic surfactant to make the NSD and studying their stability by UV-Vis (decrease of SPR peak) and DLS (size distribution and z potential) in Figure 7a, b, c.
- The total silver content in the formulation should be quantified using Inductively Coupled Plasma Mass Spectrometry (ICP-MS) or Inductively Coupled Plasma Optical Emission Spectrometry (ICP-OES) to ensure consistency in the formulation.
Response 13: We thank the recommendation about silver concentration in NSD. Please, find in supplementary material the results by MP-AES.
- The manuscript inconsistently uses the terms "AgNPs," "colloidal silver," and "silver nanoparticles." A standardized nomenclature should be adopted for clarity.
Response 14: We standardized the terms. We deleted “colloidal silver” and “silver nanoparticles” are only in the first mention text.
- Given the potential environmental risks of AgNPs, the manuscript should briefly discuss toxicity and biodegradability concerns, particularly regarding environmental accumulation.
Response 15: We appreciate your suggestion to address the potential environmental risks of AgNPs. However, the focus of our manuscript is on the health-related aspects of NSD formulations. Discussing environmental risks such as biodegradability and environmental accumulation would extend the aims and scope of this study.
- The conclusion does not outline future research directions, particularly regarding experimental validation of antiviral activity and in vivo safety studies. A discussion on the next steps would enhance the manuscript’s impact.
Response 16: We followed your recommendations and the conclusion of revised manuscript now include future directions.
L540-544: Further research, including preclinical and clinical tests, is necessary to validate the safety, efficacy, and antiviral properties of our formulation. Altogether, the resulting NSD demonstrated the desired characteristics for nasal administration and efficient coverage upon atomization. Far more important, our findings indicate that this NSD shows promise as a potential nasal spray formulation.
- The similarity score of 28% is relatively high and raises concerns about possible text overlap with existing literature. The authors must reduce this similarity score to avoid potential ethical concerns during peer review.
Response 17: Thank you to review for plagiarism our manuscript; however, we performed a plagiarism check using Grammarly, and the result indicated only 2% similarity, primarily limited to the Materials and Methods section, where some overlap with standard phrasing is expected. We have attached the plagiarism report for your reference.

Reviewer 2 Report
Comments and Suggestions for Authors
Authors proposed a paper entitled “Nasal Spray Disinfectant for Respiratory Infections Based on Functionalized Silver Nanoparticles: A Physicochemical and Docking Approach” for publication in Nanomaterials, mdpi.
The paper has good scientific soundness, and it deserves to be published after major revisions.
Introduction could be reinforced by a larger number of references.
Lines 34-86: Lack of direct comparison with existing nasal spray disinfectants. Include references to existing AgNP-based nasal formulations or other antiviral nasal sprays (e.g., carrageenan-based products) and explain how this formulation differs.
Line 81. “we developed antiviral” I would not use personal constructions, but impersonal pronouns are generally well accepted. Same as “We proposed”
Lines 80-86: the definitions of the aims of this paper could be better indicated in this paragraph.
Lines 87-102: Methodology section; in these lines there is a lack of clear experimental controls and insufficient discussion of potential limitations. Authors should include a control group (e.g., uncoated AgNPs or placebo nasal spray) to verify the effectiveness of functionalized silver nanoparticles. Also, a discussion on limitations, such as sample size constraints or potential safety concerns, should be provided.
Line 146. Even if amber was used, was this storage intended also in “dark” conditions?
Lines 186-189: Discussion of physicochemical properties; no mention of toxicity or biocompatibility testing. Authors should include cytotoxicity data (e.g., cell viability assays) to confirm the safety of AgNPs in nasal mucosa applications.
Lines 186-246: figure legends: Legends should specify conditions (e.g., solvent used in FTIR, magnification in SEM).
Lines 221-227 vs. Lines 192-198: Discrepancy between particle size measurements from different techniques. Authors should address the difference in sizes due to different measurement conditions (hydrated state in DLS vs. dry state in SEM). If necessary, reconcile results with additional techniques like TEM.
Lines 266-274: Docking results. Molecular docking binding affinity values should be compared to known inhibitors. Compare binding energies with FDA-approved antiviral compounds to assess the relative efficacy of TA-SC functionalized AgNPs.
Lines 431-444: Overstatement of AgNPs’ potential without clinical data. Authors may consider the phrase “can potentially prevent respiratory infections”, that should be revised to “shows promise as a potential nasal spray formulation, warranting further preclinical and clinical validation.”
Comments on the Quality of English LanguageA quite good use of English.
Author Response
Authors proposed a paper entitled “Nasal Spray Disinfectant for Respiratory Infections Based on Functionalized Silver Nanoparticles: A Physicochemical and Docking Approach” for publication in Nanomaterials, mdpi.
- The paper has good scientific soundness, and it deserves to be published after major revisions.
Response 1: Thank you very much for your comments and reviews.
- Introduction could be reinforced by a larger number of references.
Response 2: Introduction in manuscript is now reinforced by the following references.
Nakayama, T.; Lee, I. T.; Jiang, S.; Matter, M. S.; Yan, C. H.; Overdevest, J. B.; Wu, C.-T.; Goltsev, Y.; Shih, L.-C.; Liao, C.-K.; Zhu, B.; Bai, Y.; Lidsky, P.; Xiao, Y.; Zarabanda, D.; Yang, A.; Easwaran, M.; Schürch, C. M.; Chu, P.; Chen, H.; Stalder, A. K.; McIlwain, D. R.; Borchard, N. A.; Gall, P. A.; Dholakia, S. S.; Le, W.; Xu, L.; Tai, C.-J.; Yeh, T.-H.; Erickson-Direnzo, E.; Duran, J. M.; Mertz, K. D.; Hwang, P. H.; Haslbauer, J. D.; Jackson, P. K.; Menter, T.; Andino, R.; Canoll, P. D.; DeConde, A. S.; Patel, Z. M.; Tzankov, A.; Nolan, G. P.; Nayak, J. V. Determinants of SARS-CoV-2 Entry and Replication in Airway Mucosal Tissue and Susceptibility in Smokers. Cell Reports Medicine 2021, 2 (10), 100421. https://doi.org/10.1016/j.xcrm.2021.100421.
Harrison, A. G.; Lin, T.; Wang, P. Mechanisms of SARS-CoV-2 Transmission and Pathogenesis. Trends in Immunology 2020, 41 (12), 1100–1115. https://doi.org/10.1016/j.it.2020.10.004.
Ivanova, N.; Sotirova, Y.; Gavrailov, G.; Nikolova, K.; Andonova, V. Advances in the Prophylaxis of Respiratory Infections by the Nasal and the Oromucosal Route: Relevance to the Fight with the SARS-CoV-2 Pandemic. Pharmaceutics 2022, 14 (3), 530. https://doi.org/10.3390/pharmaceutics14030530.
Leung, N. H. L. Transmissibility and Transmission of Respiratory Viruses. Nat Rev Microbiol 2021, 19 (8), 528–545. https://doi.org/10.1038/s41579-021-00535-6.
Clementino, A. R.; Pellegrini, G.; Banella, S.; Colombo, G.; Cantù, L.; Sonvico, F.; Del Favero, E. Structure and Fate of Nanoparticles Designed for the Nasal Delivery of Poorly Soluble Drugs. Mol. Pharmaceutics 2021, 18 (8), 3132–3146. https://doi.org/10.1021/acs.molpharmaceut.1c00366.
Ibarra-Sánchez, L. Á.; Gámez-Méndez, A.; Martínez-Ruiz, M.; Nájera-Martínez, E. F.; Morales-Flores, B. A.; Melchor-Martínez, E. M.; Sosa-Hernández, J. E.; Parra-Saldívar, R.; Iqbal, H. M. N. Nanostructures for Drug Delivery in Respiratory Diseases Therapeutics: Revision of Current Trends and Its Comparative Analysis. Journal of Drug Delivery Science and Technology 2022, 70, 103219. https://doi.org/10.1016/j.jddst.2022.103219.
Chung, S.; Peters, J. M.; Detyniecki, K.; Tatum, W.; Rabinowicz, A. L.; Carrazana, E. The Nose Has It: Opportunities and Challenges for Intranasal Drug Administration for Neurologic Conditions Including Seizure Clusters. Epilepsy & Behavior Reports 2023, 21, 100581. https://doi.org/10.1016/j.ebr.2022.100581.
Illum, L. Nasal Drug Delivery—Possibilities, Problems and Solutions. Journal of Controlled Release 2003, 87 (1–3), 187–198. https://doi.org/10.1016/S0168-3659(02)00363-2.
Martín-Faivre, L.; Prince, L.; Cornu, C.; Villeret, B.; Sanchez-Guzman, D.; Rouzet, F.; Sallenave, J.-M.; Garcia-Verdugo, I. Pulmonary Delivery of Silver Nanoparticles Prevents Influenza Infection by Recruiting and Activating Lymphoid Cells. Biomaterials 2025, 312, 122721. https://doi.org/10.1016/j.biomaterials.2024.122721.
Wieler, L.; Vittos, O.; Mukherjee, N.; Sarkar, S. Reduction in the COVID-19 Pneumonia Case Fatality Rate by Silver Nanoparticles: A Randomized Case Study. Heliyon 2023, 9 (3), e14419. https://doi.org/10.1016/j.heliyon.2023.e14419.
He, Q.; Lu, J.; Liu, N.; Lu, W.; Li, Y.; Shang, C.; Li, X.; Hu, L.; Jiang, G. Antiviral Properties of Silver Nanoparticles against SARS-CoV-2: Effects of Surface Coating and Particle Size. Nanomaterials 2022, 12 (6), 990. https://doi.org/10.3390/nano12060990.
- Lines 34-86: Lack of direct comparison with existing nasal spray disinfectants. Include references to existing AgNP-based nasal formulations or other antiviral nasal sprays (e.g., carrageenan-based products) and explain how this formulation differs.
Response 3: We appreciate your suggestion. We have highlighted the comparison with existing antiviral nasal sprays in the Introduction and Discussion section.
L63-67: For example, N-palmitoyl-N-monomethyl-N,N-dimethyl-N,N,N-trimethyl-6-O gly-colchitosan, Iota-carrageenan, and hydroxypropyl methylcellulose are some antivirals that have been shown to have the appropriate physicochemical and antimicrobial characteristics for the prophylaxis of COVID-19 using the nasal route [20-23].
L 494-510: Other proposals for spray nasal disinfectants emphasize the coverage area, seek-ing to increase it by adding polysaccharides in the formulation. For instance, Moakes et al. [37] formulated a spray disinfectant to prevent COVID-19, whereas the active ingredient was Iota-carrageenan. Furthermore, the addition of gellan gum (another poly-saccharide) showed increased coverage area for all the tested ratios. Considering a 5% coverage using iota-carrageenan and 25% coverage for a 50:50 gellan gum : Iota-carrageenan ratio. Using the same active ingredients, Robinson et al. [78] improved the coverage and mucoadhesion of their formulation with the addition of low acyl gellan gum. The largest coverage area was obtained with a combination of 0.25 % Iota-carrageenan and 0.25 % low acyl gellan gum (%w/v), illustrating a 20 cm2 coated area. It is important to highlight that, for our work, the coverage protocols were similar to those of Robinson et al., resulting in higher coverage for our NSD formulations (115 cm2) and without T80 (85 cm2). These results propose the potential exploration and use of several non-ionic surfactants since polysaccharide additives have generally been used to improve absorption to the nasal mucosa due to the antimicrobial activity that some may present. Nonetheless, polysaccharides could act as a detrimental plat-form for promoting several microbial growing activities, thus negatively engraving the antimicrobial requirements of the active deliver components.
- Line 81. “we developed antiviral” I would not use personal constructions, but impersonal pronouns are generally well accepted. Same as “We proposed”
Response 4: Thank you for your suggestion. Now, the text has “we proposed”
- Lines 80-86: the definitions of the aims of this paper could be better indicated in this paragraph.
Response 5: We have improved the aims of our paper in the revised manuscript.
L80-86: Considering the above stated information and the important current efforts required to reduce or eradicate potential viral respiratory infections, here we proposed antiviral, natural based AgNPs engineered with a non-ionic ethoxylated and biocompatible surfactant coating. We proposed to assess the fine-tuned AgNPs for nasal mucous airway capable to interact with different variants of SARS-CoV-2 in accordance to our molecular docking analyses and physicochemical approach. Thus, this NSD could potentially prevent respiratory infections in an affordable and non-invasive ad-ministration way.
- Lines 87-102: Methodology section; in these lines there is a lack of clear experimental controls and insufficient discussion of potential limitations. Authors should include a control group (e.g., uncoated AgNPs or placebo nasal spray) to verify the effectiveness of functionalized silver nanoparticles. Also, a discussion on limitations, such as sample size constraints or potential safety concerns, should be provided.
Response 6: We have included limitations and further research in conclusions.
L540-544: Further research, including preclinical and clinical tests, is necessary to validate the safety, efficacy, and antiviral properties of our formulation. Altogether, the resulting NSD demonstrated the desired characteristics for nasal administration and efficient coverage upon atomization. Far more important, our findings indicate that this NSD shows promise as a potential nasal spray formulation.
- Line 146. Even if amber was used, was this storage intended also in “dark” conditions?
Response 7: Thank you very much for your observation. Now, we clarified the dark conditions in manuscript.
- Lines 186-189: Discussion of physicochemical properties; no mention of toxicity or biocompatibility testing. Authors should include cytotoxicity data (e.g., cell viability assays) to confirm the safety of AgNPs in nasal mucosa applications.
Response 8: We include cytotoxicity test and MTT in the revised manuscript.
3.4. Cell culture and cytotoxicity test
L148-172: To analyze the cytotoxicity effect of the AgNPs, we performed the MTT (3-(4,5-dimethylthiazol-2-yl)-2,5-diphenyl tetrazolium bromide) viability test, as de-scribed previously [37,38]. For the viability test we used human osteosarcoma derived osteoblast-like cell line (MG-63, ATCC CRL-1427), cultured at 1 104 cells/mL in com-plete medium constituted of Dulbecco’s modified Eagle medium (DMEM, Thermo Fisher Scientific, USA) supplemented with 10% heat-inactivated fetal bovine serum (FBS, Thermo Fisher Scientific, USA) and 100 units/mL of penicillin-streptomycin (Thermo Fisher Scientific, USA) in separate 96-well flat-bottom polystyrene culture plates (Corning, USA) at 37°C in a humidified 5% CO2 incubator for 24h. Then, the cells were washed thrice for 5 min with warm 1 × phosphate buffered saline (PBS) and incubated for 24h with serial dilutions (62.5 µg/mL - 0.4 µg/mL) of the experimental AgNPs suspended in complete medium. The AgNPs doses were discarded, and the cells were carefully washed thrice with warm PBS, to remove any artifact present in the cultures. Afterwards, the MTT protocol was carried out by adding 100 µL of MTT (5 mg/mL, Sigma Aldrich, USA) to each well of the cultured 96-well plate and incubated at 37°C in a humidified 5% CO2 incubator for 3h. The resulting formazan crystals were dissolved by removing the remaining medium containing MTT. The plate was placed in an orbital shaker at 140 rpm with 200 µL of dimethyl sulfoxide (Sigma Aldrich, USA) for 20 min. Next, the optical density (O.D.) of the dissolved crystals was meas-ured at 590 nm using a microplate reader (Thermoskan, Thermo Fisher Scientific, USA). The baseline controls were measured in a series of culture wells containing AgNPs at the experimental dilutions and prepared for MTT without MG63 as de-scribed. The negative control of cytotoxicity was selected using MG63 in complete me-dium without any treatment. Furthermore, the dose at which 50% of MG63 growth viability is inhibited (IC50) was calculated by means of a nonlinear regression curve fit using GraphPad Prism 7.03.
L300-306: The MTT assay showed an IC50 value of 17.25 µg/mL for the AgNPs used in the NSD formulation, indicating that cell viability decreases only at higher concentrations. At lower concentrations (below IC50), the treated cells exhibited minimal morpholog-ical changes and maintained viability, as illustrated in Figure 5. These results suggest that the AgNP concentrations used are below the cytotoxic threshold, confirming their safety for human cells while preserving the antimicrobial efficacy of the formulation.
- Lines 186-246: figure legends: Legends should specify conditions (e.g., solvent used in FTIR, magnification in SEM).
Response 9: We have improved Figure legends in revised manuscript.
- Lines 221-227 vs. Lines 192-198: Discrepancy between particle size measurements from different techniques. Authors should address the difference in sizes due to different measurement conditions (hydrated state in DLS vs. dry state in SEM). If necessary, reconcile results with additional techniques like TEM.
Response 10: L 278-282: A slight discrepancy between the particle sizes measured by DLS and SEM is observed; however, this difference is expected due to the sample preparation and measurement conditions. It means an aqueous colloidal solution for DLS and dried for SEM.
- Lines 266-274: Docking results. Molecular docking binding affinity values should be compared to known inhibitors. Compare binding energies with FDA-approved antiviral compounds to assess the relative efficacy of TA-SC functionalized AgNPs.
Response 11: We appreciate your recommendation. We have included a comparison between FDA-approved antibodies for COVID-19 and our NSD.
L428-438: Comparing our binding energy results with FDA-approved antibodies for COVID-19 prophylaxis (tixagevimab, pemivibart, and cilgavimab), only one study has reported the molecular docking and binding energies of tixagevimab and cilgavimab on the receptor-binding domain (RBD) of the SARS-CoV-2 spike protein for the Delta variant [70,71]. The reported values in this study were -13.9 kcal/mol for tixagevimab and -12.4 kcal/mol for cilgavimab. In contrast, the binding energy of the TA-SC complex in our study for the Delta variant was -8.22 kcal/mol. While this difference is notable, it suggests that the TA-SC complex could still compete in binding affinity. Interestingly, we obtained stronger binding energies for the wild type (-10.05 kcal/mol) and Omicron (-9.62 kcal/mol) variants, indicating that the TA-SC complex presents a more promising prophylactic potential against these variants than Delta.
- Lines 431-444: Overstatement of AgNPs’ potential without clinical data. Authors may consider the phrase “can potentially prevent respiratory infections”, that should be revised to “shows promise as a potential nasal spray formulation, warranting further preclinical and clinical validation.”
Response 12: Thank you for your suggestion. Now, the text has “NSD shows promise as a potential nasal spray formulation, warranting further pre-clinical and clinical validation”

Reviewer 3 Report
Comments and Suggestions for Authors
The paper is devoted for nasal spray disinfectant for respiratory infections based on functionalized silver nanoparticles: a physicochemical and docking approach. The topic is generally interesting, however the paper contain unexplained places (below) and need major revisions.
Line 173 the term Kollman charges should be explained.
Line 187-189 and 204-209 the corresponding references should be added.
Results presented in Fig. 5 should be more discussed. Particularly, the difference between AgNPs and NSD properties should be explained.
Figs. 7 – 10 should be more commented.
Conclusions should be rewritten in more informative way.
Author Response
The paper is devoted for nasal spray disinfectant for respiratory infections based on functionalized silver nanoparticles: a physicochemical and docking approach. The topic is generally interesting, however the paper contains unexplained places (below) and need major revisions.
- Line 173 the term Kollman charges should be explained.
Response 1: Thank you for your observation. We extended the explanation of Kollman charges in the revised manuscript.
L: Briefly, Kollman charges which represents the charge distribution of complex molecules and hydrogens were added, and water molecules were removed and saved in pdbqt format.
- Line 187-189 and 204-209 the corresponding references should be added.
Response 2: We have enriched the text with new references.
Line 187-189
Amirjani, A.; Haghshenas, D. F. Ag Nanostructures as the Surface Plasmon Resonance (SPR)˗based Sensors: A Mechanistic Study with an Emphasis on Heavy Metallic Ions Detection. Sensors and Actuators B: Chemical 2018, 273, 1768–1779. https://doi.org/10.1016/j.snb.2018.07.089.
La Spina, R.; Mehn, D.; Fumagalli, F.; Holland, M.; Reniero, F.; Rossi, F.; Gilliland, D. Synthesis of Citrate-Stabilized Silver Nanoparticles Modified by Thermal and pH Preconditioned Tannic Acid. Nanomaterials 2020, 10 (10), 2031. https://doi.org/10.3390/nano10102031.
References included in Line 204-209
Ranoszek-Soliwoda, K.; Tomaszewska, E.; Socha, E.; Krzyczmonik, P.; Ignaczak, A.; Orlowski, P.; Krzyzowska, M.; Celichowski, G.; Grobelny, J. The Role of Tannic Acid and Sodium Citrate in the Synthesis of Silver Nanoparticles. J Nanopart Res 2017, 19 (8), 273. https://doi.org/10.1007/s11051-017-3973-9.
Ricci, A.; Olejar, K. J.; Parpinello, G. P.; Kilmartin, P. A.; Versari, A. Application of Fourier Transform Infrared (FTIR) Spectroscopy in the Characterization of Tannins. Applied Spectroscopy Reviews 2015, 50 (5), 407–442. https://doi.org/10.1080/05704928.2014.1000461.
(32) Husanu, E.; Chiappe, C.; Bernardini, A.; Cappello, V.; Gemmi, M. Synthesis of Colloidal Ag Nanoparticles with Citrate Based Ionic Liquids as Reducing and Capping Agents. Colloids and Surfaces A: Physicochemical and Engineering Aspects 2018, 538, 506–512. https://doi.org/10.1016/j.colsurfa.2017.11.033.
- Results presented in Fig. 5 should be more discussed. Particularly, the difference between AgNPs and NSD properties should be explained.
Response 3: We have highlighted the difference between AgNPs and NSD in the following lines:
L338-344: Despite the similar behavior of AgNPs and NSD, indicating that the addition of T80 does not significantly alter the colloidal system as expected, it's important to note the difference in absorbance. The higher absorbance of NSD compared to AgNPs suggests a more homogeneous particle size distribution and minimized Ag oxidation, which can be attributed to the key role of T80 in preventing aggregation and preserving the opti-cal properties of AgNPs during storage.
- 7 – 10 should be more commented.
Response 4: We have extended the discussion of molecular docking results comparing with FDA-approved antibodies.
L385-395: Comparing our binding energy results with FDA-approved antibodies for COVID-19 prophylaxis (tixagevimab, pemivibart, and cilgavimab), only one study has reported the molecular docking and binding energies of tixagevimab and cilgavimab on the receptor-binding domain (RBD) of the SARS-CoV-2 spike protein for the Delta variant [70,71]. The reported values in this study were -13.9 kcal/mol for tixagevimab and -12.4 kcal/mol for cilgavimab. In contrast, the binding energy of the TA-SC complex in our study for the Delta variant was -8.22 kcal/mol. While this difference is no-table, it suggests that the TA-SC complex could still compete in binding affinity. Interestingly, we obtained stronger binding energies for the wild-type (-10.05 kcal/mol) and Omicron (-9.62 kcal/mol) variants, indicating that the TA-SC complex presents a more promising prophylactic potential against these variants than Delta.
- Conclusions should be rewritten in more informative way.
Response 5: Thank you very much for your recommendation. The revised manuscript includes new conclusion in more informative way.
L527-544: Nasal sprays offer a promising alternative to conventional drug administration routes such as oral, intramuscular, transcutaneous methods, particularly for assessing the respiratory pathway. However, it is crucial to consider key physicochemical factors that facilitate crossing this biological barrier. In this study, we developed a NSD colloidal based solution that fulfilled the synthesis of spherical AgNPs with uniform size distribution (<20 nm), a negative surface charge, and high stability (<-30 mV). The enhancement of stability and mucoadhesion was reached by the formation of a tannic acid – sodium citrate complexing with the AgNPs and the addition of T80 a nonionic surfactant.
Moreover, molecular docking results evidenced prominent binding interactions between TA-SC stabilized AgNPs and key receptors in ACE2 and the SARS-CoV-2 spike protein from main variants (Wuhan, Delta, and Omicron). These findings suggest that NSD is not only suitable for nasal administration but also has significant potential in providing efficient coverage and sustained delivery to the respiratory tract.
Further research, including preclinical and clinical tests, is necessary to validate the safety, efficacy, and antiviral properties of our formulation. Altogether, the resulting NSD demonstrated the desired characteristics for nasal administration and efficient coverage upon atomization. Far more important, our findings indicate that this NSD shows promise as a potential nasal spray formulation.

Round 2
Reviewer 1 Report
Comments and Suggestions for Authors
Although I initially recommended rejection due to major scientific and structural shortcomings, after carefully revision, the authors have satisfactorily addressed most of the concerns raised in the previous round. The revised manuscript presents a significantly improved version of the experimental work, with more comprehensive discussions, better-structured content, and clearer justification of the methodologies and interpretations. Furthermore, the authors provided a reasonable explanation for the overlap percentage, attributing it to standard methodological descriptions and properly cited content. Given the substantial improvements in both scientific quality and presentation, I now consider the manuscript to be acceptable for publication in its current form.
Reviewer 2 Report
Comments and Suggestions for Authors
Authors addressed my issues point by point.
Reviewer 3 Report
Comments and Suggestions for Authors
Authors make proper corrections according to reviewer remarks and I suggest publish the paper as it is.